



# A novel spectroscopic approach and sampling method for ambient hydrogen chloride detection: HCl-TILDAS

John W. Halfacre[1], Jordan Stewart[1], Scott C. Herndon[2], Joseph R. Roscioli[2], Christoph Dyroff[2], Tara I. Yacovitch[2], Michael Flynn[3], Stephen J. Andrews[1], Steven S. Brown[4,5], Patrick R. Veres[4], Pete M. Edwards[1]

[1]Wolfson Atmospheric Chemistry Laboratories, Department of Chemistry, University of York, Heslington, York, YO10 5DD, UK
[2]Aerodyne Research, Inc., Billerica, MA, 01821, USA
[3]Department of Earth and Environmental Science, Centre for Atmospheric Science, School of Natural Sciences, The University of Manchester, Manchester M13 9PL, UK
[4]Chemical Sciences Laboratory, National Oceanic and Atmospheric Administration, Boulder, CO, 80305, USA
[5]Department of Chemistry, University of Colorado, Boulder, CO 80309, USA

*Correspondence to:* John Halfacre (john.halfacre@york.ac.uk)*,* Pete Edwards (pete.edwards@york.ac.uk)

**Abstract.** The largest inorganic, gas phase reservoir of chlorine atoms in the atmosphere is hydrogen chloride (HCl), but the challenges in quantitative sampling of this compound cause difficulties for obtaining high-quality, high-frequency measurements. In this work, tunable infrared laser direct absorption spectroscopy (TILDAS) was demonstrated to be a superior optical method for sensitive, in situ detection of HCl at the 2925.89645 cm$^{-1}$ absorption line using a 3 $\mu$m interband cascade laser. The instrument has an effective path length of 204 m, 1 Hz precision of 7-8 pptv, and 3$\sigma$ limit of detection ranging from 21-24 pptv. For longer averaging times, the highest precision obtained was 0.5 pptv and 3$\sigma$ limit of detection of 1.6 pptv at 2.4 minutes. HCl TILDAS was also shown to have high accuracy when compared with a certified gas cylinder, yielding a linear slope within the expected 5% tolerance of the reported cylinder concentration (slope = 0.964 ± 0.008). The use of heated inlet lines and active chemical passivation greatly improve the instrument response times to changes in HCl mixing ratios, with minimum 90% response times ranging from 1.2 - 4.4 s, depending on inlet flow rate. However, these response times lengthened at relative humidities > 50%, conditions under which HCl concentration standards were found to elicit a significantly lower response (-5.8%). The addition of high concentrations of gas phase nitric acid (> 4.0 ppbv) were found to increase HCl signal (< 10%), likely due to acid displacement with HCl or particulate chloride adsorbed to inlet surfaces. The equilibrium model ISORROPIA suggested a potential of particulate chloride partitioning into HCl gas within the heated inlet system if allowed to thermally equilibrate, but field results did not demonstrate a clear relationship between particulate chloride and HCl signal obtained with a denuder installed on the inlet.

## 1 Introduction

Growing attention is being given to the role of reactive chlorine in tropospheric oxidation chemistry (Simpson et al., 2015), given its potential impacts on the lifetimes of volatile organic compounds; atomic chlorine reacts with hydrocarbons at rate constants often orders of magnitude greater than those with hydroxyl radical (Burkholder et al., 2015; Atkinson et al., 2006; Jahn et al., 2021), as in Reaction (R1), where R represents an alkane:

$$RH + Cl^{\bullet} \rightarrow HCl + R^{\bullet} \tag{R1}$$



Even moderate amounts of such a potent oxidizer could lead to changes in concentrations of $O_3$, $NO_x$, and hydroxyl
radicals. However, the high reactivity of atomic chlorine radicals, combined with a lack of effective gas phase
recycling mechanisms, only allows for a small degree of accumulation, with global tropospheric averages
estimated to range between $10^2$-$10^5$ atoms $cm^{-3}$ (Allan et al., 2001; Pszenny et al., 2007; Wang et al., 2021;
Wingenter et al., 1996; Singh et al., 1996). As such, in situ, quantitative detection of atomic chlorine radicals
remains out of reach. It is instead more practical to study chlorine through relatively more abundant and stable
reservoir species, such as hydrogen chloride (e.g., Angelucci et al., 2021), molecular chlorine (e.g., Liao et al.,
2014), chlorine monoxide (e.g., Tuckermann et al., 1997), and nitryl chloride (e.g., Osthoff et al., 2008).

48        Hydrogen chloride (HCl) is of particular interest because it is the most abundant form of inorganic

chlorine in the gas phase and acts as both a source and end-product of atomic chlorine. Reaction (R1) represents
a significant gas phase HCl formation pathway, but its largest atmospheric source on a global basis is sea salt
aerosol via acid displacement (Graedel and Keene, 1995, 1996; Wang et al., 2019; Erickson et al., 1999), in which
the presence or uptake of other acids, such as nitric acid ($HNO_3$) or even organic acids (Laskin et al., 2012), shifts
the equilibrium of aqueous chloride back toward gas phase HCl, as in Reaction (R2) (Brimblecombe and Clegg,
1988; Clegg and Brimblecombe, 1986):

$HX(g) \leftrightarrow H^+ (aq) + X^- (aq)$                                    (R2)

Additional contributions to the HCl budget come from volcanic emissions (von Glasow et al., 2009; Graedel and
Keene, 1996) and anthropogenic emissions, including coal combustion, biomass burning, industrial processes
(e.g., smelting, cement production), and solid waste incineration (Zhang et al., 2022; Fu et al., 2018; Keene et al.,
1999; McCulloch et al., 1999; Ren et al., 2017; Wang et al., 2019). The loss processes for HCl are governed by
two major sinks: reaction with hydroxyl radical and deposition. The reaction of HCl with hydroxyl radical in
Reaction (R3) directly produces chlorine radicals that can participate in tropospheric oxidation, but is relatively
slow ($k = 7.8$ x $10^{-13}$ $cm^3$ $molecule^{-1}$ $s^{-1}$ at 298 K) (Atkinson et al., 2007):

$HCl + {}^{\bullet}OH \rightarrow Cl^{\bullet} + H_2O$                                          (R3)

While deposition of HCl removes a chlorine atom from the gas phase, its eventual uptake into an aqueous solution
will produce chloride ions that can be reintroduced into the atmosphere, either by deacidification (as in R2), or
via oxidation into other volatile molecular halogens (i.e., $Cl_2$, ICl, BrCl) (Abbatt et al., 2010; Fickert et al., 1999;
Frinak and Abbatt, 2006; Knipping et al., 2000; Oum et al., 1998) or nitryl chloride (Behnke and Zetzsch, 1990;
Behnke et al., 1997, 1992). Recent field observations and modelling suggest the vast majority of tropospheric
HCl can be found within 1 km of the surface, with mixing ratios decreasing with height until reaching the
tropopause, where mixing ratios begin increasing again (Wang et al., 2019, 2021; Lee et al., 2018; Haskins et al.,
2018). In the lower troposphere, ambient HCl mixing ratios are typically observed between $10^1$ and $10^3$ parts per
trillion by volume (pptv), with the highest amounts found in polluted, coastal regions (Angelucci et al., 2021;
Crisp et al., 2014, and references therein; Tao et al., 2022).

78        Recent technological advances have enabled the production of suitable instrumentation for online, in situ

detection of ambient HCl. Chemical ionisation mass spectrometry (CIMS) is one such method, and has been





previously characterized in laboratory studies by $3\sigma$ limits of detection as low as 15 pptv and sensitivities as high
as 2-4 counts sec$^{-1}$ pptv$^{-1}$ (Eger et al., 2019a; Marcy et al., 2004; Roberts et al., 2010). CIMS instruments are also
robust enough to deploy on mobile platforms, including aircraft (Marcy et al., 2004; Veres et al., 2008) and ships
(Eger et al., 2019b). The primary disadvantages to CIMS exist in the possibility of sampling compounds (e.g.,
water) that may interfere with the desired ionisation chemistry (e.g., Marcy et al., 2004), as well as issues of
selectivity arising from non-analytes that create signal interferences at the desired mass-to-charge ratios meant to
represent HCl and/or confirm appropriate isotopic ratios and high limits of detection (Eger et al., 2019a; Roberts
et al., 2010). Additionally, CIMS instruments can be quite heavy, require low vacuums, have high power
consumption, and often require use of large amounts of consumables (e.g., $N_2$ gas).
An alternative, well-understood approach for HCl detection is infrared absorption spectroscopy. Optical
methods benefit from analysing well-defined and spectrally isolated HCl absorption features (Toth et al., 1970;
Li et al., 2011), resulting in a virtually absolute and specific measurement technique. Previously published
literature for laser-based HCl instrumentation has demonstrated potential efficacy for in situ detection, including
cavity-enhanced (Wilkerson et al., 2021; Hagen et al., 2014; Furlani et al., 2021) and multi-pass cells (Harris et
al., 1992; Webster et al., 1994; Scott et al., 1999), both of which benefit from path lengths spanning hundreds of
meters to kilometers. These instruments have also been tested on mobile platforms, such as ships (Harris et al.,
1992), aircraft (Webster et al., 1994), and balloons (Scott et al., 1999; Wilkerson et al., 2021). The development
of small, thermoelectrically cooled, interband cascade lasers (ICLs) in recent years has increased the portability
of these instruments while also allowing the ability to probe the major HCl infrared absorption feature wavelength
($\sim$3.42 $\mu$m).
CIMS and optical methods have both proven to be excellent means of gas phase HCl detection. However,
quantitative sampling remains a challenge for all existing measurement techniques. Hydrogen chloride has a large
dipole moment and strong hydrophilocity, which makes it susceptible to interactions with polar surface groups,
or surfaces on which water may be present. This "sticky" behavior results in long instrument response times
during HCl concentration changes (e.g., > 60 seconds) under sampling configurations that include sample tubing
and particle filters (Furlani et al., 2021). Further, even inert surfaces, such as those made from
polytetrafluoroethylene (PTFE) or perfluoroalkoxy (PFA) Teflon, contain sites where HCl or other sticky
molecules (e.g., $HNO_3$) may sorb (Roscioli et al., 2016; Neuman et al., 1999; Yokelson et al., 2003); it is has also
been estimated that PFA Teflon tubing may contain water films between 0.1-10 $\mu$m thickness at 20-50% relative
humidity, which will readily interact with small polar molecules (Liu et al., 2019; Laasonen and Klein, 1997).
Several coatings have been reported in the literature to improve sticky-compound transmission, including
halocarbon wax applied to glass (Yokelson et al., 2003; Webster et al., 1994), inert silicon coatings applied to
stainless steel (Wilkerson et al., 2021), and continual flow of polyfluorinated acid vapor across glass and Teflon
(Roscioli et al., 2016).
In this work, we present a novel optical method for the detection of HCl: Tunable Laser Infrared Direct
Absorption Spectroscopy (TILDAS), combined with a sampling methodology to minimise inlet artefacts. The
TILDAS technique has the advantage of being highly sensitive due to its 204 m pathlength, a fast response time
via incorporation of "active passivation," and being virtually specific for HCl.



## 2 Materials and experimental methods

### 2.1 Gases and Chemicals

For in-lab experiments, dry air for sample background measurements was generated with an air compressor and dehumidifying system (dew point approximately -60º C, absolute water vapor concentration ~0.01%). When testing the effects of water on the sampling configuration in the laboratory, air was manually humidified using a Michell Instruments DG-3 Dewpoint Generator. This compressed air system was also used in generating nitrogen ($N_2$) gas with a commercial $N_2$ generator (Infinity NM32L, Peak Scientific Instruments, United Kingdom), which was used as carrier gas for active passivation (Sect. 2.3) and permeation sources (Sect. 2.4). During field studies, zero-grade air (270028-L, BOC Limited, United Kingdom) and oxygen-free $N_2$ (44-W, BOC Limited, United Kingdom) were used for these purposes (Sect. 2.5).

Perfluorobutanesulfonic acid (PFBS, 97% purity, CAS 375-73-5, Sigma Aldrich, United States) was used to actively chemically passivate inlet surfaces (Sect. 2.3). Concentrated HCl solution (37% HCl, CAS 7647-01-0, Fisher Scientific, United States) and concentrated nitric acid ($HNO_3$) solution (70%, CAS 7697-37-2, Fisher Scientific, United States) were used in making permeation source standards (Sect. 2.4). A 5 ppm HCl gas cylinder (diluted in $N_2$, certified as 4.7 ppm ± 5%, 2760716, BOC Limited, United Kingdom) was used as an independent method validation standard (Sect. 2.4).

### 2.2 HCl-TILDAS

#### 2.2.1 TILDAS Design

The HCl-TILDAS instrument was developed at and purchased from Aerodyne Research Inc (McManus et al., 2011, 2015). The underlying principle of the tunable infrared laser direct absorption spectrometry (TILDAS) technique is infrared absorption spectroscopy. Briefly, light from a 3µm-interband cascade laser (operated at 24.03ºC) is collected by an objective, and then is focused through a flip-in pinhole, removed during sampling. After this focus, the beam is reimaged into the multi-pass, astigmatic Herriott cell. In addition, a beam splitter enables the laser to travel down a reference path used intermittently to measure and verify the laser tuning rate. The Herriott cell used in this instrument has an effective path length of 204 m, and is held to a temperature of 29 ºC by circulating air past temperature controlled liquid along the sides of the instrument (Oasis Model T-Three). Temperature controlling the interior of the TILDAS mitigates the effects of exterior temperature changes that may cause optical fringe effects in the reported mixing ratios or changes to the mirror and table distances that may affect the path travelled by the laser light reaching the detector.

This incident radiation probes the strong R(1) $H^{35}Cl$ line (2925.89645 cm$^{-1}$) of the (1-0) rovibrational absorption band near 3.4 $\mu$m (Guelachvili et al., 1981). The instrument software sweeps the laser over the desired spectral window, which it can find via strong absorption lines from other spectrally close absorbers, including methane (2926.18 cm$^{-1}$, 2926.700231 cm$^{-1}$) and water (2926.456 cm$^{-1}$, 2926.742 cm$^{-1}$). In addition, the laser is coincidentally able to estimate concentrations of methanol (2925.851 cm$^{-1}$, 2925.998 cm$^{-1}$), formaldehyde (2925.842 cm$^{-1}$, 2926.1 cm$^{-1}$), and nitrogen dioxide (2925.8 cm$^{-1}$, 2926.128 cm$^{-1}$). Since the absorbing features in this region are well-resolved, spectral interferences for HCl are not expected for typical ambient mixing ratios observed for the above species.





### 2.2.2 Sampling Inlet

Filtration of particulate matter is required to protect and maintain the efficacy of the multi-pass optics described in the previous section (McManus et al., 1995), as well as reduce the potential of scattering and absorption from particulates within the cell. However, traditional paper filters and filter holders provide surfaces onto which HCl may be removed from the sample stream, both lowering the observed concentration and providing a reservoir of HCl that could be later forced back into the gas phase via acid displacement. To obviate this problem, a custom-fabricated quartz virtual impactor (hereafter referred to as "inertial inlet") was added into the instrument sampling line (Fig. 1). The inertial inlet glass is housed within a temperature-controlled enclosure set to 50 °C (Omega CNi32). Sample air passes from an ambient pressure region through a critical orifice into a low-pressure region (< 100 torr). The resulting flow rate through the instrument was determined by the size of the critical orifice in the inertial inlet and cell pressure (set to approximately 40 torr); because different inlets were used for these experiments, flow rates were 2.8, 3.7 or 12.7 L min$^{-1}$, yielding cell residence times ($1/e$) of 2.0 s and 1.5 s, and 0.4 s respectively. Once in the low-pressure region, large particles (> 300 nm diameter) have large forward momentum and travel straight into a waste flow path (approximately 13% of the total volumetric flow). Meanwhile, gas molecules and particles with an approximate diameter < 300 nm have less inertia and can make the 180° turn necessary to continue along the sample flow path into the TILDAS (approximately 87% of the total volumetric flow); because the astigmatic Herriott cell used in the TILDAS has a shorter path length / higher light throughput than high finesse cavity systems, it is not as sensitive to decreased light throughput caused by the accumulation of smaller diameter particulate matter on cell mirrors. The inertial inlet is connected to the HCl-TILDAS via 3m of insulated, temperature controlled (50 °C), 3/8" PFA Teflon tubing.

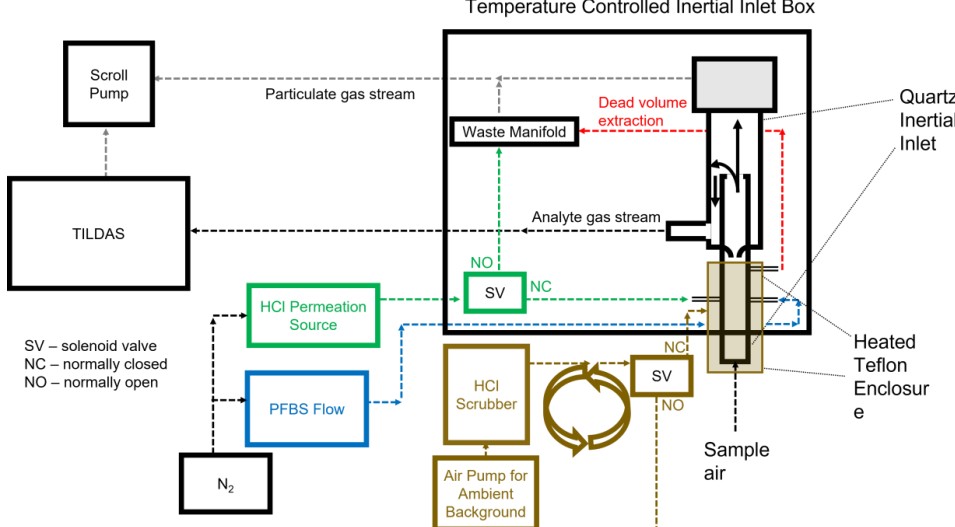

**Figure 1: Experimental flow schematic for sampling HCl**

### 2.3 Active Passivation

It has been previously shown that adding a small, continuous flow of PFBS vapor to sampling lines is effective at increasing transmission of HNO$_3$ through sampling tubing (Roscioli et al., 2016). This technique was used in this work to minimize loss of HCl to surfaces between the inertial inlet and the optical cell. Five-mL of PFBS was


contained within a bubbler (made from perfluoroalkoxy (PFA) Teflon or Pyrex for laboratory and field studies,
respectively). Compressed $N_2$ gas was passed into the bubbler to flush the headspace (containing PFBS vapor)
into the analyte flow path, just after the point of sample air entry into the inertial inlet (Fig. 1). Addition of fresh
PFBS vapor into the flow path may quickly release several ppbv of HCl from unpassivated surfaces and may take
several hours to finish conditioning the system. The temperature and carrier gas flow rate (containing PFBS) were
adjusted (between 18-22 °C and 50-100 mL min$^{-1}$, respectively) until no additional HCl was released to ensure
optimal passivation conditions. Given the growing body evidence on the deleterious effects of perfluorinated
compound accumulation in the environment (Buck et al., 2011), release of PFBS vapor was mitigated by adding
a scrubber containing hydroxide salts, glass wool, and activated charcoal to the pump exhaust. When replacement
was necessary, the bubbler and any contaminated tubing were washed with absolute ethanol and fully dried before
re-use, with rinsings collected and disposed of as hazardous waste.
Passivation efficacy was regularly tested as a function of the timescale of signal change resulting from
the addition / removal of HCl standard flow into the inertial inlet (Fig. 1). Timescales were calculated as detailed
in Sect. 2.6.2.

**2.4 HCl Standards for Technique Validation**

Custom HCl permeation sources were created for regular inlet transmission testing using a method modified from
Furlani et al. (2021). HCl was pipetted into a 2” length of PTFE tubing (0.118” ID, 0.157” OD, VWR). Tubing
was sealed by heating the ends, one at a time, in a small flame until the tubing became transparent. The end of
the tubing was then clamped by pliers and removed from the flame, creating a seal on cooling. The completed
permeation source was then placed in a temperature-controlled aluminum block (set to 35 °C). A flow (30 mL
min$^{-1}$) of $N_2$ gas, carries the HCl vapor into the instrument flow path (Fig. 1). Additionally, a permeation source
for $HNO_3$ was created and utilized in the same manner for the purposes of studying interferences (Sect. 3.3.2).
A cylinder of 5 ppmv (4.7 ± 5%) HCl (Sect. 2.1) was used to confirm both the TILDAS response to HCl,
as well as the permeation source output. On opening the cylinder for the first time (or after a period of disuse),
multiple days of constant flow (controlled between 1-50 mL min$^{-1}$ by an Alicat MCS-50SCCM) were required to
condition the regulator before HCl-TILDAS reflected a stable output. Because TILDAS is an optical method that
relies on characteristic, well-described absorption features of molecules, it is considered an absolute detection
method and does not require frequent calibrations.

**2.5 Field Testing**

To demonstrate its performance as an in situ, field-ready instrument, the HCl-TILDAS was deployed during the
Integrated Research Observation System for Clean Air (OSCA) campaign at the University of Manchester
(Manchester, United Kingdom, approximately 53.444 °N, 2.216 °W), and sampled HCl between 10 June - 22 July,
2021. The OSCA campaign seeks to understand and assess urban air pollution and air quality at various sites
across the UK in order to inform and support policy makers in making future decisions, as well as evaluating the
impacts of decisions previously made. More information on the campaign and links to relevant studies can be
found here: https://gtr.ukri.org/projects?ref=NE%2FT001917%2F1#/tabOverview. The measurement site was
located at the Manchester Air Quality Super Site on the Firs Environmental Research Station at the University of





Manchester campus, and sampled air masses are believed to be heavily influenced by the surrounding urban
environment.
The TILDAS instrument and pump for generating background measurements (KNF Model
N035.1.2AN.18) were installed within an air-conditioned shipping container, held at 25 °C.  The inertial inlet,
HCl permeation source, and active passivation unit were integrated into a separate box (80 cm x 60 cm), installed
above the container roof (~ 3m AGL) (Fig. 2).  Because each of these components are operated at different
temperatures (inertial inlet box, permeation source, and active passivant held at 50, 35, and 18 °C, respectively),
the larger box was cooled with a water-cooling fan (controlled to 25 °C) to buffer the box interior from changes
in the external ambient temperatures and direct solar heating.  Temperatures were regularly checked using
thermocouples interfaced with an Arduino Uno (Arduino).

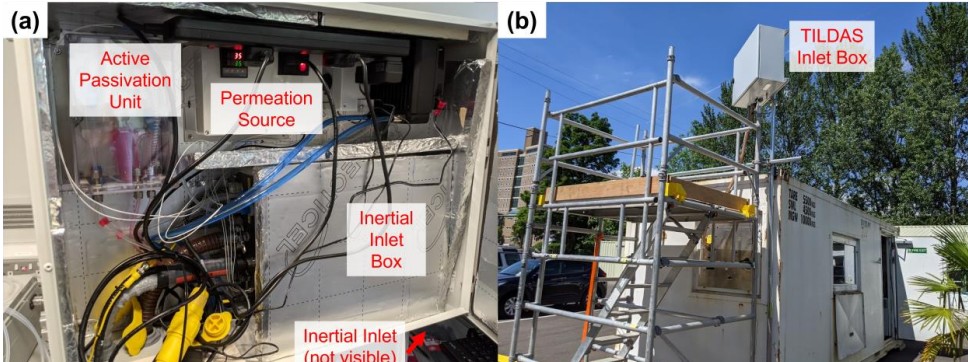

**Figure 2: a) Field configuration for HCl TILDAS inlet system.  b) Mounted inlet system at Manchester field site.**
During the campaign, blank measurements were obtained for 2 min out of every 10 min throughout
ambient sampling periods in order to check for drifts in instrument background signal due to optical stability.  An
effective blank was achieved by passing ambient air through a trap composed of activated charcoal and glass
wool.  This HCl-scrubbed air was then directed to a Teflon encasing around the inertial inlet, which then
overflowed the inlet at approximately 35 L min$^{-1}$, such that the inlet would only be sampling scrubbed air.  To
evaluate the inlet for losses and the efficacy of the PFBS, flow from the HCl permeation source was added directly
into the inertial inlet on top of the background air overflow for 9 min every 3 hr.  Note that overblows using zero
air cylinders were found to cause a large increase in HCl signal, followed by a slow decay; it is believed this is
due to the sudden disruption in the equilibrium of water molecules adsorbed to instrumentation surfaces.  For this
reason, permeation source additions under dry air conditions were performed overnight when ambient HCl
chemistry mixing ratios were believed to be low.  For these experiments, compressed dry air (produced by Jun
Air OF302-25MQ2) overflowed the inlet for 1 hr, and permeation source HCl was added across three 10-min
intervals within this hour.
**2.6 Data Analysis**
Data processing for this work, including background corrections and uncertainty analysis, were conducted
primarily using the *R* statistical software (R Core Team, 2021) in tandem with the RStudio environment (RStudio
Team, 2021).


### 2.6.1 Background Correction


As discussed above, background measurements were obtained for 2 min out of every 10 min sampling period.
The median of the final 30s of each background period was used as an offset value. Offset values between these
points were estimated by linear interpolation and were subsequently subtracted from ambient observations for
analysis.

### 2.6.2 HCl Signal Response Timescales


Timescales of signal decay ($\tau$) after removal of a HCl standard (Sect. 2.4) from the HCl-TILDAS sampling line
were calculated as an objective measure of the sampling method performance. Such timescales for sticky gases
(including HCl) have been previously determined by fitting data to a biexponential model (Roscioli et al., 2016;
Zahniser et al., 1995; Ellis et al., 2010; Pollack et al., 2019):
$$y = A_1 exp\left(-\frac{t}{\tau_1}\right) + A_2 exp\left(-\frac{t}{\tau_2}\right) \qquad\qquad (1)$$

where $y$ represents the HCl mixing ratio, $t$ represents elapsed time, both $A_1$ and $A_2$ are proportionality terms, and
both $\tau_1$ and $\tau_2$ control the shape of the decay curve. Herein, both single exponential and biexponential models
were fit to the data to determine the time needed to reach $1/e$ ($\tau$), 75% ($\tau_{75}$), and 90% ($\tau_{90}$) of a starting HCl
concentration. The fitting function within $R$ (i.e., "nls") required initial guesses for the $A$ and $\tau$ terms, which were
based on the starting mixing ratio of HCl and anticipated residence time of air in the absorption cell, respectively;
however. the function was not constrained to these values in formulating its output.

### 2.6.3 HCl Partitioning


The thermodynamic equilibrium model ISORROPIA II (Fountoukis and Nenes, 2007), used to investigate $K^+$–
$Ca^{2+}$–$Mg^{2+}$–$NH_4^+$–$Na^+$–$SO_4^{2-}$–$NO_3^-$–$Cl^-$–$H_2O$ aerosol systems, was employed to estimate the potential that
particulate chloride (pCl$^-$) may partition to HCl within the heated inlet system. Calculations were performed in
'forward mode' when possible, in which the total (gas + aerosol) concentrations of $NH_3$, $H_2SO_4$, HCl, $HNO_3$, $Na^+$,
$Ca^{2+}$, $K^+$, and $Mg^{2+}$ were specified, alongside ambient temperatures and relative humidities. The model then solves
a series of equilibrium equations based on these conditions, incorporating water activity equations, activity
coefficient calculations, electroneutrality, and mass conservation, to determine the gas and aerosol concentrations
at thermodynamic equilibrium. The calculations were then repeated for different potential TILDAS sample line
testing temperatures (35, 50 and 80°C) to determine changes in gaseous HCl mixing ratios resulting from re-
partition with aerosols within the sample line. In scenarios where gas phase concentrations were unknown, the
model was initialised in 'reverse mode' with averaged aerosol concentrations to predict gas phase concentrations
at equilibrium. In all model calculations, the aerosol was assumed to be in a thermodynamically stable state, in
which salts precipitate if saturation is exceeded, owing to the low relative humidities within the heated inlet line.

### 3 Results & Discussion


### 3.1 Instrument Performance


The performance metrics of HCl-TILDAS are compared with previously described optical methods in Table 1.
Allan-Werle deviations were calculated in the laboratory while overflowing the inlet with dry zero air (Sect 2.5)



(Hagen et al., 2014; Furlani et al., 2021), and in the field with HCl-scrubbed sample air (i.e., without removal of
water vapor) (Fig. 3). Under 30s integration times and using the 3.7 L min$^{-1}$ inlet, the precision (1-2 pptv at $1\sigma$)
and $3\sigma$ limit of detection (4-6 pptv) outperform previously reported methods, which range from 6 - 100 pptv
precision, and 18 - 78 pptv limits of detection under 30 s averaging times. HCl-TILDAS has clear advantages for
both figures of merit if longer integration times are considered; for dry, laboratory conditions, we achieved a
precision of 0.5 pptv and corresponding LOD of 1.6 pptv at the Allan minimum of 2.4 minutes, compared with
1.5 pptv precision and 4.4 pptv LOD for field observations at an Allan minimum of 56 seconds. These values are
more than adequate for obtaining high quality field observations at the expected ambient HCl mixing ratios of
$10^1$-$10^3$ pptv (Wang et al., 2019).

**Table 1: Summary table comparing the performance of HCl TILDAS to similar, previously reported optical methods.**
**[a] The lower limit of the figures of merit represent laboratory sampling, while the higher limit represents field sampling.**
**$\tau_{90}$ are reported for dry, laboratory sampling conditions. The lower value represents laboratory analysis, while the**
**higher value represents data from field work (Fig. 9). [b] Reported for mixing ratio changes > "$10^9$ per volume or higher".**

| Instrument | LOD | Precision | $\tau_{90}$ | Reference |
|---|---|---|---|---|
| HCl-TILDAS[a] | 21-24 pptv (1 s)<br>4-6 pptv (30 s) | 7-8 pptv (1 s)<br>1-2 pptv (30 s) | > 4.4 (±0.3) s (2.8 slpm)<br>> 1.15 (±0.06) s (12.7 slpm) | This study |
| Near-IR CRDS | < 18 pptv (30 s) | 6 pptv (30 s) | > 10 s | Furlani et al. (2021) |
| Near-IR CRDS | 60 pptv (60 s) | 20 pptv (60 s) | 10 - 15 s | Hagen et al. (2014) |
| Off-axis integrated cavity output spectrometer (OA-ICOS) | 78 pptv (30 s) | 26 pptv (30 s) | 10 s | Wilkerson et al. (2021) |
| Aircraft laser infrared absorption spectrometer (ALIAS) | 33 pptv (30 s) | 100 pptv (30 s) | 10 s[b] | Webster et al. (1994) |






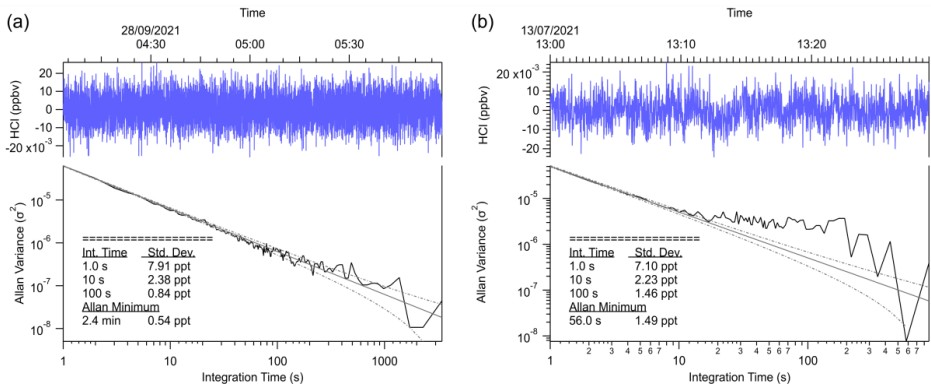

**Figure 3: Allan variance plot demonstrating the signal variance and limit of detection calculations for varying integration times.**

A commercial HCl cylinder with a certified concentration (4.7 ppm ± 5%) was used as an objective standard for in-lab validation. Mixing ratios were varied by adjusting the flow rate of the cylinder output, which was then directly injected into an inertial inlet sidearm (Fig. 1) for direct injection into the passivated inertial inlet. Standard HCl was then diluted into the dry, HCl-free compressed air being sampled by TILDAS. The slope obtained (0.964) was found to lie within the expected 5% uncertainty, reflecting high accuracy for TILDAS observations (Fig. 4). However, additional sources of error causing deviation from unity must be considered. For example, multiple days of HCl cylinder flow are required for the output mixing ratio to stabilize at its maximum concentration (as observed by TILDAS) after opening the cylinder; this behavior is presumably caused by uptake of HCl onto the metal cylinder regulator and Teflon tubing lines until they are fully conditioned, causing the observed signal to register lower than expected. Changes to HCl cylinder flow additionally require similar conditioning time to re-establish signal stability, likely caused by changes to the HCl gas/surface equilibrium.

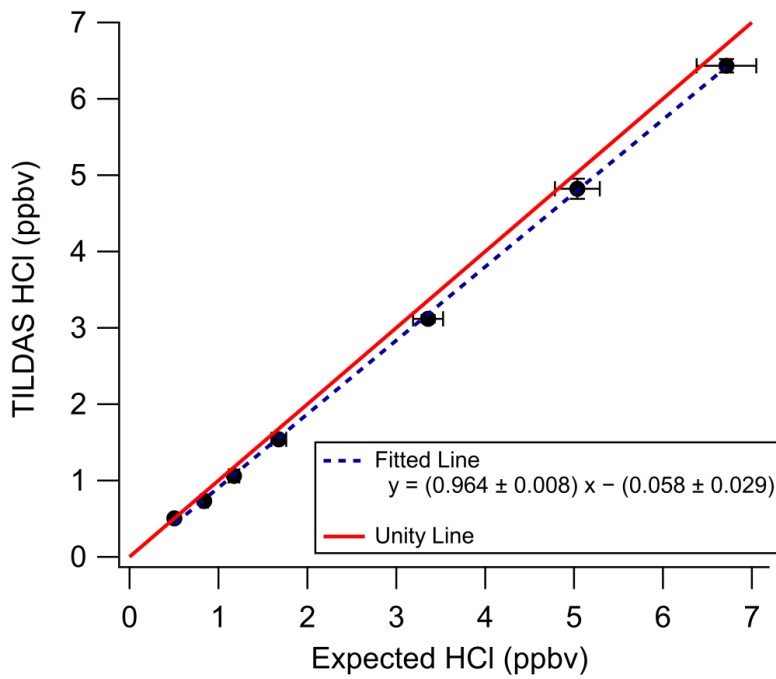

**Figure 4: In-lab validation of HCl-TILDAS by a commercial HCl standard. Principal axis error bars represent the 5% uncertainty associated with the HCl standard (as reported by the manufacturer), while the vertical axis error bars represent 1 standard deviation of HCl-TILDAS observations for each validation point.**

### 3.2 Evaluation of Sampling Method

Multiple variables were found to affect HCl transmission through the instrument flow path (Fig. 1), including the presence or absence of active passivation (i.e., whether PFBS is flowing through the sample line; Sect 3.2.1) and the presence of water vapor (Sect 3.2.2). The timescales of signal change after removal of an HCl source were used to objectively compare the relative effects of each variable. They further allow for direct comparison of the performance of this HCl sampling method with those previously published (Table 1). Note that these timescales reflect how quickly HCl mixing ratios change within the 1.8L measurement cell and do not include the time required for the sample gas to reach the cell (i.e., time zero is when a change in signal is first observed, not from when an addition valve was triggered).

### 3.2.1 Active Passivation

To test the effectiveness of active passivation, HCl permeation source flow was added into the TILDAS sample line for 10 min of subsequent 30 min periods using the inertial inlet with the lowest flow rate (2.8 L min$^{-1}$), as the effects of HCl-wall interactions would be the most exaggerated. Experiments were repeated both with and without the coinciding flow of PFBS (Fig. 5), and the TILDAS inlet was overflowed with dry, compressed air (Sect. 2.1), such that a baseline signal was observed in the absence of permeation source addition. As seen in Fig. 5a, employing active passivation yields sharp, square wave-like behavior on both addition and removal of the HCl permeation source flow. From the fits of a single exponential model, $\tau_e$ averaged $1.9 \pm 0.2$ s (N = 21) after HCl



permeation source removal (Fig. 5b, Fig. A1), which compares well with the predicted absorption cell residence
time ($1/e$) of 2.0 s. Though a biexponential model was also fit to these data (and achieved comparable $\tau_e$, $\tau_{75}$, and
$\tau_{90}$ values, Table A1), the convergence tolerance of the non-linear least squares solving algorithm (Sect. 2.6.2)
had to be loosened by six orders of magnitude (from a default value of $1 \times 10^{-5}$ to $2 \times 10^{1}$) to achieve convergence,
suggesting these results are not meaningful. Indeed, the errors for the predicted variables often greatly exceeded
the magnitude of the associated variables themselves, suggesting a biexponential model is not appropriate for
these actively passivated data.

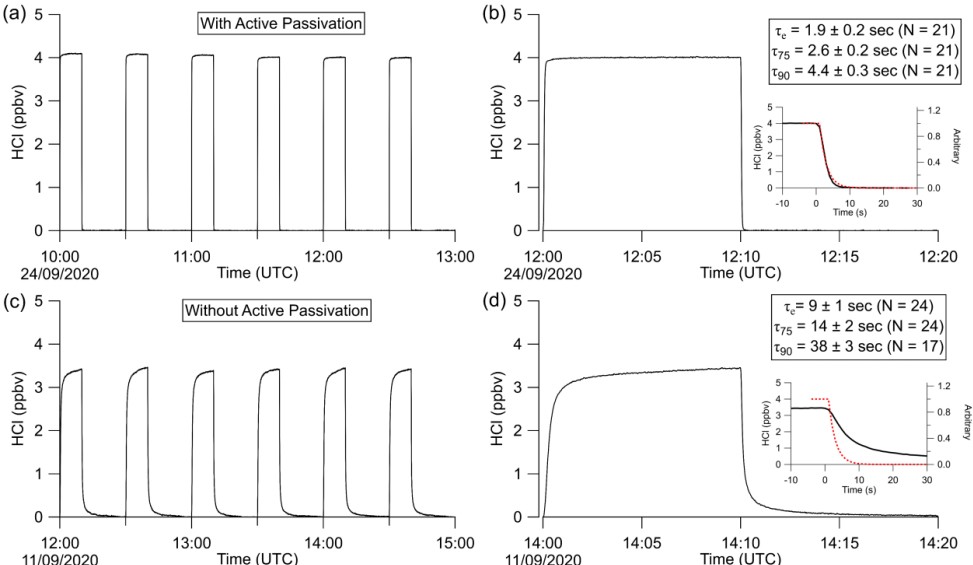

**Figure 5: Excerpted time series of HCl permeation source additions with (a, b) and without (c, d) use of active**
**passivation. a) TILDAS response to HCl permeation source addition to the sample line for 10 minutes every 30 minutes.**
**b) Example case from plot (a) demonstrating the profile of the decay timescales. Reported $\tau$'s represents the mean and**
**standard deviation of the entirety of these experiments. Inset shows a close-up of the actual decay compared with the**
**red dashed line, representing the theoretical decay profile of a non-sticky compound modelled on the residence time of**
**air in the absorption cell. Frames c) and d) are analogous to a) and b), but without use of active passivation.**
Without active passivation, the signal profiles of the HCl additions have comparatively slower rises, and
do not reach the average HCl maximum mixing ratios of $4.03 \pm 0.06$ ppbv within 10 min intervals (Fig. 5a, b, Fig.
A2). In these cases, biexponential models were fit without having to adjust the default convergence tolerance,
and the results were found to have smaller term- and residual errors when compared to the analogous single
exponential model (see Table A2). $\tau_e$ for the signal decays was calculated as $9 \pm 1$ s (N = 24), or approximately
4.5 times greater than the residence time through the measurement cell (Fig. 5c, d).
The reported timescales in this work can be further improved by increasing the flow rate of the inlet.
Using the 12.7 slpm inertial inlet, $\tau_e$ averaged $0.49 \pm 0.03$ s (N=21), comparing very well to the theoretical cell
residence time ($1/e$) of 0.45 s for this flow rate (Fig. 6, Fig. A3, Table A3). $\tau_{90}$ was similarly improved, averaging
$1.15 \pm 0.06$ s. The higher flow rate clearly demonstrates that wall interactions are reduced; as demonstrated by
Fig. 6, the decay rate mimics that of methane, which is a non-sticky compound also measured by the HCl-TILDAS.
As the current configuration includes 3 m of heated tubing between the inertial inlet and the HCl-TILDAS itself,
it is likely this response could be further improved by shortening this line.

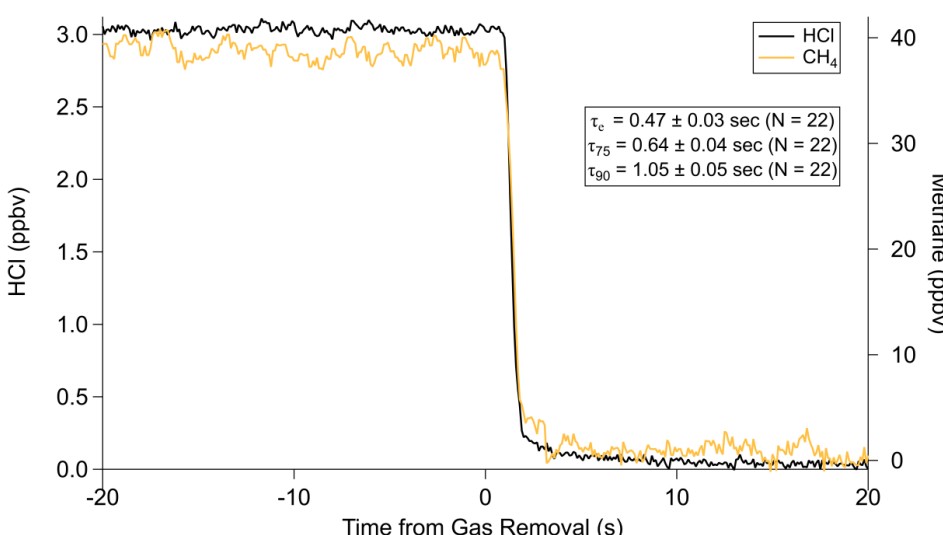

**Figure 6: Comparison of HCl decay with methane at inlet flow rate of 12.7 slpm.**

The $\tau_{90}$ achieved utilizing active passivation in this study is the shortest reported instrument response time for changes in HCl mixing ratios to date (Table 1) and demonstrates that the use of PFBS is effective for reducing HCl-surface interactions. Previous studies have suggested that a biexponential model (Eq. 1) may better physically represent sticky gas flow through an instrument (Furlani et al., 2021; Zahniser et al., 1995; Ellis et al., 2010; Pollack et al., 2019); in this approach, $\tau_1$ may represent the air residence time within the instrument, while $\tau_2$ will represent the factor(s) that cause the analyte to lag through the instrument (e.g., surface interactions). Our results were not inconsistent with this postulation since the unpassivated cases were well-represented by the biexponential model (i.e., significant $\tau_1$ and $\tau_2$ equation terms within Eq. 1), while passivated cases were better represented by the single exponential model (i.e., dominant $\tau_1$ but negligible $\tau_2$). However, the results do not directly support it either; for unpassivated cases, the predicted $\tau_1$ averaged 6.2 ± 0.7 (greater than 3 times the cell residence time for the inertial inlet used), and 69 ± 10 for $\tau_2$ (Table A2). Further reconciliation of the physical basis behind the biexponential model is outside the scope of this work, and no attempt is made to ascribe further physical meaning to the derived coefficients.

**3.2.2 Humidity**

The experiments in the previous sections were conducted using dried compressed air. As dry air is not representative of ambient sampling conditions, timescale experiments were also performed with humidified sample air under passivated conditions. The results in Fig. 7a demonstrate a clear increase in $\tau$'s with increasing relative humidity, affecting $\tau_{90}$ most prominently.





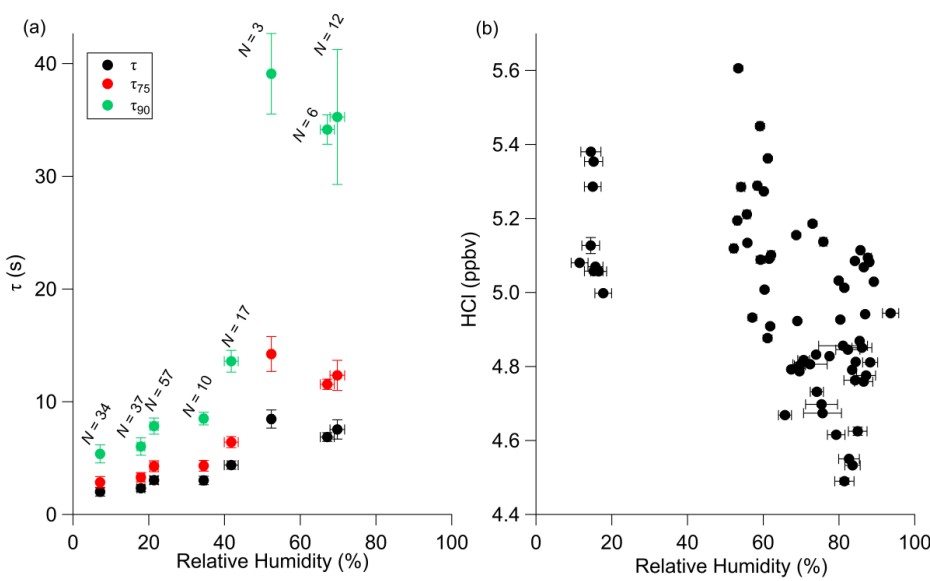

**Figure 7: Effects of changes in relative humidity on a) $\tau$ in laboratory experiments and b) HCl standard mixing ratios in the field. Relative humidity values are based on the TILDAS-observed water mixing ratio observed during the HCl decay period (a) or HCl standard addition (b), and concurrent temperature reading. Error bars for both axes represent one standard deviation.**

While Roscioli et al. (2016) note that the general effectiveness of active passivation on HNO₃ instrument response times appeared independent of humidity levels between 0-70%, the results of this experiment do not display this same behavior above approximately 40% relative humidity. Notably, the inlet flow rate used for these experiments is less than four times that used in that study (i.e., 2.8 L min⁻¹ vs 14 L min⁻¹), which would increase analyte-surface interactions. However, these values do represent an improvement from the HCl sampling method reported by Furlani et al. (2021), in which $\tau_{90}$ was reported as 239s at 33% relative humidity. Similarly, field additions of a HCl permeation source (utilizing the 3.7 L min⁻¹ inertial inlet) elicited lower mixing ratios at relative humidities above 60% (mean of $4.9 \pm 0.2$ ppbv), contrasting with additions under dry air conditions (i.e., relative humidities below 20%; mean of $5.2 \pm 0.1$ ppbv) (Fig. 7b). This finding suggests that a permanent or semi-permanent physical loss of HCl is occurring within the sampling inlet at higher humidities, resulting in an average -5.8% bias. Both PFA tubing and silica surfaces have been previously reported to adsorb several monolayers-worth of water at room temperature in humid air (Saliba et al., 2001; Sumner et al., 2004), which would be expected to bind and solvate HCl. As both the inertial inlet and sample line were heated to 50 °C, it is anticipated that this effect would be minimised by discouraging water from attaching to surfaces, but not eliminated. However, increasing the sampling temperatures may further improve both the instrument response timescale and reduce this loss effect; warmer temperatures may also increase the likelihood of HCl degassing from coarse mode particles within the inertial inlet before their removal, or from fine mode particles that may travel throughout the entire sample path (Brimblecombe and Clegg, 1988). Further discussion of the effects of particulate chloride and uncertainty estimation can be found in Sect. 3.3.1.



### 3.3 Potential Interferences

As discussed above, spectral interferences are not believed to play a major role in the detected HCl concentrations. However, two potential sources of undesired HCl may exist if sample gas contains a significant amount of particulate chloride (pCl⁻) or other strong gaseous acids (e.g., $HNO_3$), discussed in more detail below.

### 3.3.1 Effects of Particulate Chloride

It is well established that HCl and particulate chloride (pCl⁻) exist together in dynamic equilibrium. The use of heated sample inlet lines (50 °C in this study) may volatilize HCl from pCl⁻ if sufficient heating occurs before particles are removed via impaction, yielding measurements with positive systematic error. As discussed in Sect. 2.6.3, the thermodynamic equilibrium model ISORROPIA II was used to theoretically assess the impact of pCl⁻ volatilisation within the heated TILDAS sample inlet on measured HCl mixing ratios based on three potential operating temperatures (35°C, 50°C, and 80°C). To simulate conditions of an inland, urban environment, averaged aerosol concentrations from London, England, were used to initiate the model (Crilley et al., 2017; Bandy et al., 2022b, a). It was estimated for the conditions of these measurements that HCl repartitioning from pCl⁻ would result in an increase of the HCl mixing ratio by 1 ppqv at both 35°C and 50 °C, while dramatically increasing to 200 pptv at 80°C. Such increases in HCl are expected to derive from the loss of the liquid aerosol phase following the reduction in humidity experienced in the elevated temperatures of the sample inlet, and the evaporation of $NH_4Cl$. However, Huffman et al. (2009) reported approximately the evaporation of only 10-15% $NH_4Cl$ aerosol through a thermodenuder held at 50 °C (12 s residence time). Based on an inertial inlet flow rate of 2.8 L min⁻¹ and a corresponding residence time of 150 ms before particulate removal via impaction, it is unlikely volatilization will significantly affect these measurements. Further in situ testing was performed during the OSCA field study, discussed further in Sect. 3.4.

### 3.3.2 Effects of Nitric Acid

The use of PFBS appears to lessen the effects of HCl surface adsorption and improve the instrument response time to changes in HCl concentrations (Fig. 5, 6). If, though, PFBS does not completely prevent HCl sorbing to walls, the sampling of acids stronger than HCl (e.g., $HNO_3$) may perturb the existing passivation equilibrium on instrument surfaces. In order to test this, a $HNO_3$ permeation source was fabricated (Sect. 2.4) and allowed to flow into the TILDAS inlet (Fig. 8). The $HNO_3$ permeation source output was estimated as NO using a Mo-catalyzed $NO_y$ convertor in tandem with a commercial $NO_x$ analyzer (Teledyne T200). In a test experiment, the addition of 4.0 ppbv of $HNO_3$ to the inertial inlet caused a maximum increase of 0.29 ppbv to the HCl signal (Fig. 8). Continued addition of $HNO_3$ eventually causes the signal to plateau at a higher background, ~0.08 ppbv above the original background.





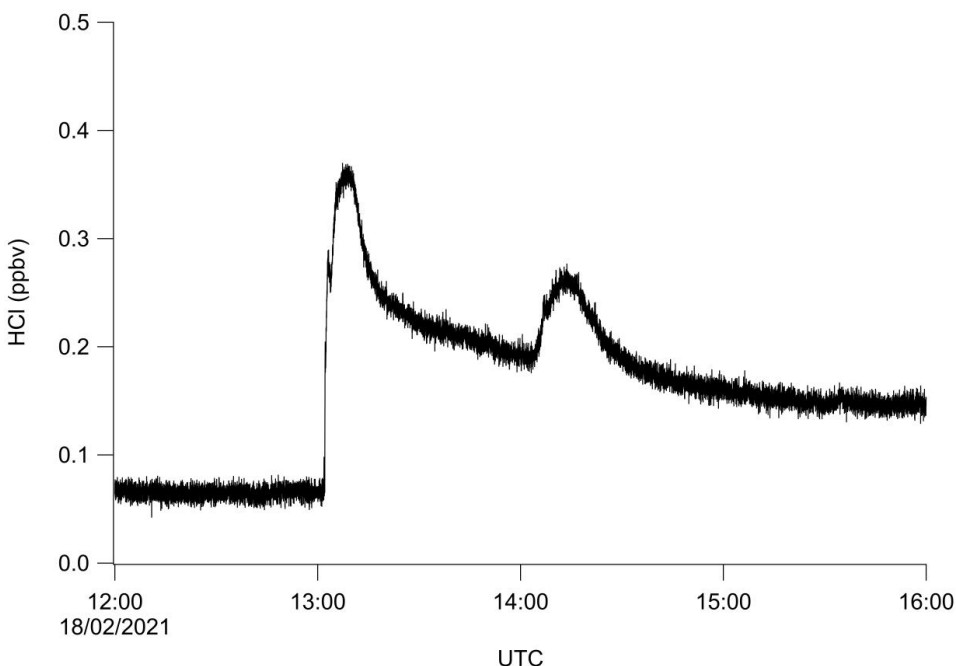

**Figure 8: Demonstration of the effects of 4.0 ppbv nitric acid addition to the passivated sample inlet flow at approximately 13:00 UTC.**

There is no absorption band overlap between $HNO_3$ and HCl in the analyzed spectral region, strongly indicating the observed increase in HCl signal occurred due to additional HCl molecules reaching the absorption cell. It is plausible this occurs because of interactions between $HNO_3$ and surfaces where HCl may be adsorbed, or with sampled particulates. One possible mechanism is that the $HNO_3$ increases competition for sorption sites, and ultimately replaces HCl on the surface. In this scenario, expected behavior would be a gradual increase in the background HCl signal as the stronger acid removes available sorption sites, and increased HCl throughput is achieved. A second mechanism would occur if water or particulate $Cl^-$ are present on instrument surfaces; here, the diffusion of the $HNO_3$ into the water would cause acid displacement of HCl, as in Reaction (R2). If the strong acid flux were large enough, a sharp HCl signal increase (commensurate with the magnitude of available $Cl^-$) would be anticipated from HCl off-gassing that would gradually recover as a new equilibrium is established. As seen in Fig. 8, it appears that a combination of these mechanisms is present. Once equilibrium had been established with addition of $HNO_3$, flow from additional $HNO_3$ permeation sources were added to the inertial inlet to observe whether additional HCl would be driven off (results not shown). However, each addition of $HNO_3$ resulted in similar spikes and signal recoveries to elevated HCl background levels. As the sudden introduction of 4.0 ppbv $HNO_3$ into the TILDAS inlet produced < 10% of a signal response, it is likely a more gradual introduction of $HNO_3$ would elicit a proportionally smaller HCl signal. Further, Fig. 8 was produced using an inertial inlet flow rate of 2.8 L $min^{-1}$; these mechanisms are expected to be further reduced using faster-flow inlets (e.g., 12.7 L $min^{-1}$), which would both reduce gas-surface interactions, as well as make the mixing ratio transient proportionally smaller.





While this interference was shown to be of potential significance in a laboratory context, in situ effects
cannot be quantified without concurrent $HNO_3$ (or proxy) observations. To this end, estimations of how $HNO_3$
affects our method in a real-world context are further explored in Sect. 3.4.
**3.4 Field Sampling**
Field observations for HCl-TILDAS were obtained during the Summer 2021 OSCA campaign, hosted at the
University of Manchester (Sect. 2.4; Fig. 9). These represent the second high frequency tropospheric field
measurement of HCl reported by optical techniques (Angelucci et al., 2021). For the period presented, ambient
relative humidity ranged from 36-98%, and corresponded with average $\tau_e$ of $2.8 \pm 0.3$ s ($\tau_{90} = 7 \pm 1$ s). Because
the inertial inlet used in this study had a flow rate of 3.7 L min$^{-1}$, the expected $1/e$ residence time in the Herriott
cell is approximately 1.5 s; these longer empirical instrument response timescales indicate incomplete passivation
of inlet surfaces. Further, as discussed in Sect. 3.2.2, it is expected that the magnitude of the HCl measurements
will be biased low by as much as 5.8% in this campaign due to inlet surface losses, quantified through regular
field additions of a HCl permeation standard (Fig. 7).

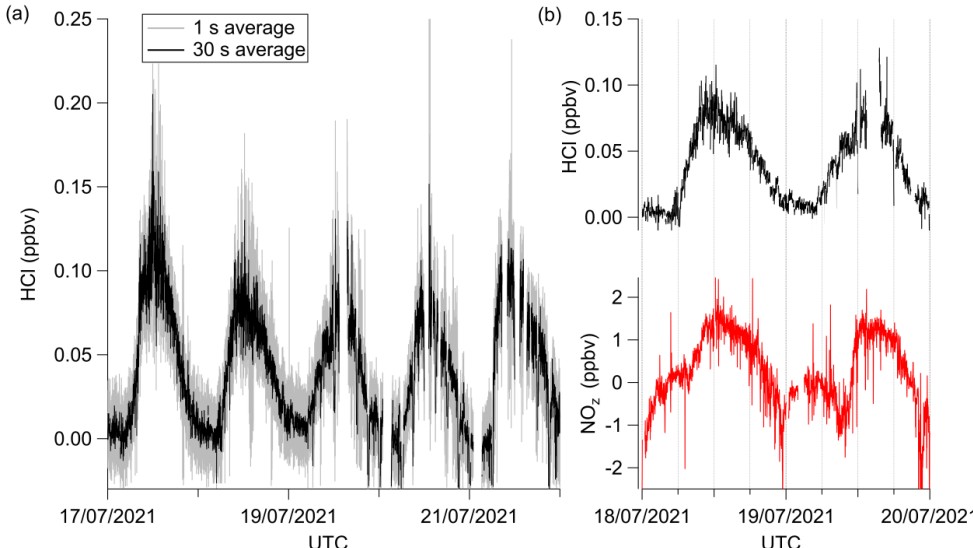

**Figure 9 – Excerpted field data from summer OSCA 2021 campaign. a) Averaged time series during the final week of**
**measurements, in which grey represents 1 s data collection frequency, while the black trace represents 30 s averages of**
**these same data. b) Comparison of HCl time series (top) and concurrent $NO_z$ time series, both averaged to 1 min.**
Additional sources of uncertainty may be introduced from plumes of $HNO_3$ sampled by our inlet, as
discussed in Sect. 3.3.2. While no direct $HNO_3$ measurement was obtained during the OSCA campaign, $NO_z$ was
used as an approximation, calculated from co-located $NO_x$ and $NO_y$ observations ($NO_z = NO_y - NO - NO_2$)
(Watson, 2022c, b). For the period presented in Fig. 9b, a Pearson correlation coefficient ($r$) of 0.69 was found
between HCl and $NO_z$. Given both compounds ambient production pathways are expected to follow a
photochemically driven diurnal cycle, this suggestion of linearity is not surprising. However, the profiles
themselves differ, with changes in $NO_z$ lagging changes in HCl. For example, HCl mixing ratios begin to rise at
06:00 on 18 July 2021, while $NO_z$ mixing ratios remain comparatively plateaued until 08:00, when it begins its





rise.  A similar pattern repeats on 19 July 2021, in which HCl mixing ratios begin rising just before 06:00, and
$NO_z$ mixing ratios again do not increase until 08:00.  The sharp increase in $NO_z$ mixing ratios after 08:00 is not
followed by an in-kind increase in HCl mixing ratios; if $HNO_3$ were eliciting HCl within the sample inlet, it would
be expected fluxes of $HNO_3$ would precede or coincide with increases in HCl.  As such, we do not believe $HNO_3$
is a significant interference within our inlet for the period analysed here.
To test the extent to which $pCl^-$ may repartition to HCl, a denuder was temporarily fitted in line to sample
only $pCl^-$; consequently, any HCl observed during the time period could be attributed to the re-partitioning of $pCl^-$
within the TILDAS sample inlet (Fig. 10).  To confirm the efficacy of removing HCl gas, cylinder additions that
result in TILDAS observed mixing ratios of 2.8, 35, and 69 ppbv were injected through the denuder for 60s with
no corresponding increase in TILDAS signal (Fig. 10c).  For the period presented, HCl signal was seen to range
between limits of detection to peaking at 53 pptv.  ISORROPIA was used to test how much HCl may originate
from $pCl^-$ in the conditions during the OSCA campaign, utilizing co-located measurements of total (gas + aerosol)
concentrations of $NH_3$ and $HNO_3$ (as $NO_z$) (Watson, 2022c, b, a) within the heated inlet system using the 'forward'
mode in ISORROPIA (no metals were included in these calculations).  Based on these simulations, it was expected
that the majority of $pCl^-$ would partition into the gas phase upon reaching thermal equilibrium in the sample inlet
leading to systematic errors of up to 40, 43, and 48 pptv at 308, 323, and 353 K respectively.  While the HCl signal
did reach these values while the denuder was installed, no direct relationship was observed between the HCl signal
and concurrent $pCl^-$ measurements (Fig. 10a, b).  In particular, there are instances (e.g., between 12:00-15:00 on
20 June 2022) where the available chlorine (calculated as the mixing ratio of chlorine if it were entirely released
from particulates) is less than HCl observations. This may suggest a potential leak between the denuder and the
inertial inlet that could allow a small volume of ambient air to contaminate the air sample, obfuscating accurate
interpretation of these results.  While a strong relationship was not observed between the $pCl^-$ and HCl signals
(with denuder) in the period observed here, the ISOROPPIA predictions emphasize that this is a significant
possible source of positive error in HCl measurements whenever heated sample lines are used for HCl sampling
in the presence of particulates.



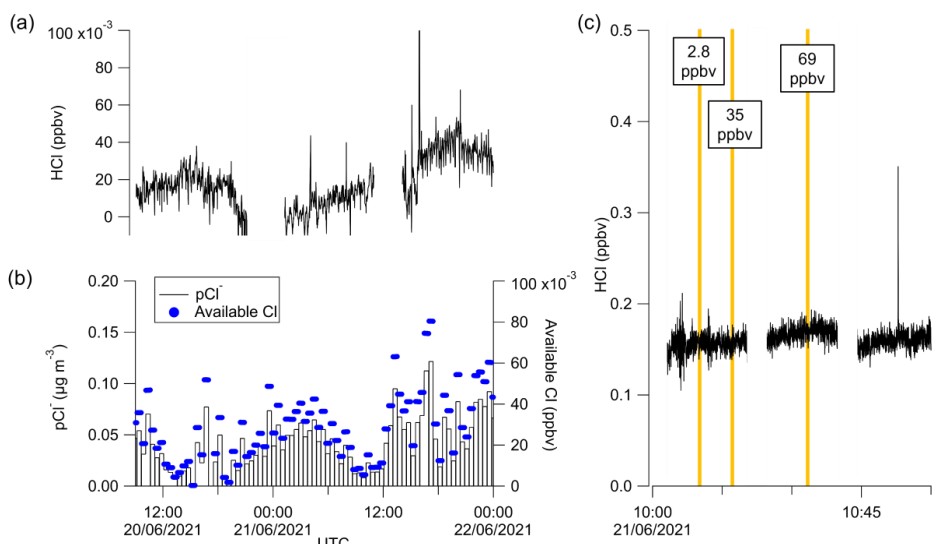

**Figure 10: Time series of a) HCl when denuder was installed on HCl TILDAS inlet in comparison with b) pCl⁻ observations. Available Cl was calculated by converting the pCl⁻ concentrations into mixing ratios. c) HCl cylinder additions were conducted (yellow shading) to verify the denuder was removing gas phase HCl. The data in panel c) has neither been background corrected or time averaged.**

## 4 Conclusions

This work has demonstrated the viability of HCl-TILDAS for obtaining high-frequency observations of ambient HCl. The associated sampling method, involving a virtual impactor to avoid excess surface-mediated interactions with filters, as well as heat and chemical passivation to increase HCl throughput, was also shown to greatly improve instrument response to changes in HCl concentration. However, there is room for further innovation in obviating the stickiness of HCl, including additional heating of sampling lines, minimizing pressure within the sampling line, as well as utilizing higher flow inlets. The use of shorter inlets operating at higher flow rates will additionally reduce sample air residence time in the inlet, both reducing HCl-wall interactions and mitigating the likelihood of HCl partitioning out of particulates within the inlet. The fast time responses to changes in HCl mixing ratios shown herein will be well-suited for mobile sampling platforms, such as aircraft or vehicle-based laboratories, in which high temporal and spatial concentration variability are inherent. Finally, the potential for interferences from particulate chloride necessitates careful consideration for the method of obtaining background measurements. Regular installations of a denuder, or incorporation of a denuder into a background mechanism would minimize the uncertainty presented.





**Appendix A**

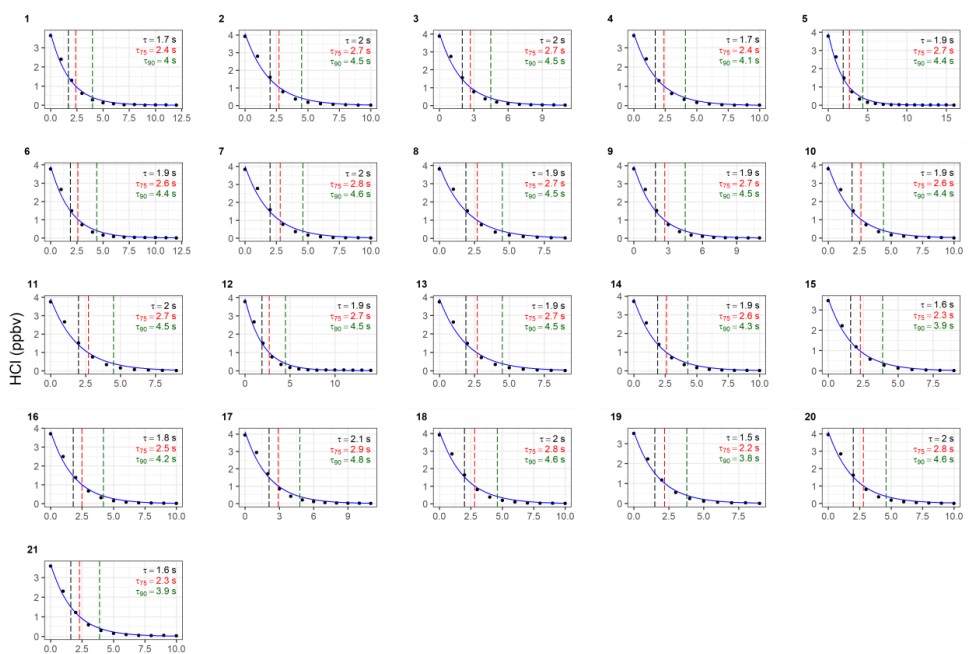

**Figure A1 – Instrument response times to changes in HCl mixing ratios utilising active passivation. Black dots**
**represent observed data and are overlayed by the calculated single exponential model (according to the terms listed in**
**Table A1). Vertical hashed lines are placed on time elapsed corresponding to $\tau_e$ (black), $\tau_{75}$ (red), and $\tau_{90}$ (green).**





**Table A1 – Results for each model fit for determining the instrument response times under actively passivated**
**conditions with the 2.8 L min$^{-1}$ inertial inlet, and corresponds with Fig. A1.   Model parameters correspond to Eq. 1 in**
**Sect. 2.6.2.**

| | | | | Single Exponential Fit | | | | | | Bi-Exponential Fit | | | | |
|---|---|---|---|---|---|---|---|---|---|---|---|---|---|---|
| Trial | $\tau_e$ (s) | $\tau_{75}$ (s) | $\tau_{90}$ (s) | $A_1$ | $k_1$ | Residuals | $\tau_e$ (s) | $\tau_{75}$ (s) | $\tau_{90}$ (s) | $A_1$ | $k_1$ | $A_2$ | $k_2$ | Residuals |
| 1 | 1.7 | 2.4 | 4 | 3.7 ± 0.1 | 0.58 ± 0.01 | 0.10 | 1.4 | 2.1 | 4 | 3.4 ± 1.2 | 0.8 ± 0.3 | 0.5 ± 1.2 | 0.2 ± 0.3 | 0.20 |
| 2 | 2 | 2.7 | 4.5 | 4.1 ± 0.1 | 0.60 ± 0.02 | 0.16 | 1.9 | 2.8 | 4.9 | 1.8 ± 10.4 | 0.8 ± 1.9 | 2.3 ± 10.4 | 0.4 ± 0.6 | 0.22 |
| 3 | 2 | 2.7 | 4.5 | 4.0 ± 0.1 | 0.60 ± 0.02 | 0.14 | 1.8 | 2.6 | 4.9 | 2.7 ± 4.1 | 0.8 ± 0.7 | 1.4 ± 4.1 | 0.3 ± 0.4 | 0.23 |
| 4 | 1.7 | 2.4 | 4.1 | 3.7 ± 0.1 | 0.58 ± 0.02 | 0.11 | 1.7 | 2.4 | 4.2 | 1.7 ± 12.1 | 0.8 ± 1.9 | 2.0 ± 12.1 | 0.4 ± 0.7 | 0.16 |
| 5 | 1.9 | 2.7 | 4.4 | 3.9 ± 0.1 | 0.60 ± 0.01 | 0.11 | 1.9 | 2.8 | 5 | 2.7 ± 4.1 | 0.7 ± 0.5 | 1.2 ± 4.2 | 0.3 ± 0.4 | 0.17 |
| 6 | 1.9 | 2.6 | 4.4 | 3.9 ± 0.1 | 0.60 ± 0.02 | 0.13 | 1.9 | 2.7 | 4.9 | 3.0 ± 4.2 | 0.6 ± 0.5 | 0.9 ± 4.2 | 0.3 ± 0.5 | 0.20 |
| 7 | 2 | 2.8 | 4.6 | 4.0 ± 0.2 | 0.60 ± 0.02 | 0.17 | 1.9 | 2.8 | 4.9 | 2.3 ± 10.3 | 0.7 ± 1.4 | 1.8 ± 10.3 | 0.3 ± 0.7 | 0.24 |
| 8 | 1.9 | 2.7 | 4.5 | 4.0 ± 0.1 | 0.60 ± 0.02 | 0.16 | 2 | 2.8 | 4.7 | 0.3 ± 21.3 | 0.8 ± 17.6 | 3.6 ± 21.3 | 0.5 ± 0.7 | 0.19 |
| 9 | 1.9 | 2.7 | 4.5 | 4.0 ± 0.1 | 0.60 ± 0.02 | 0.14 | 1.9 | 2.7 | 4.8 | 2.3 ± 5.6 | 0.8 ± 1.1 | 1.7 ± 5.6 | 0.3 ± 0.4 | 0.21 |
| 10 | 1.9 | 2.6 | 4.4 | 3.9 ± 0.1 | 0.60 ± 0.02 | 0.14 | 2 | 2.7 | 4.2 | -4.5 ± 40.0 | 1.0 ± 1.6 | 8.3 ± 40.0 | 0.7 ± 0.5 | 0.09 |
| 11 | 2 | 2.7 | 4.5 | 3.9 ± 0.2 | 0.60 ± 0.02 | 0.16 | 2 | 2.7 | 4.5 | 4.9 ± 40.5 | 0.6 ± 0.9 | -1.1 ± 40.5 | 0.8 ± 7.0 | 0.16 |
| 12 | 1.9 | 2.7 | 4.5 | 3.9 ± 0.1 | 0.60 ± 0.02 | 0.12 | 1.6 | 2.2 | 4.2 | 3.8 ± 0.7 | 0.7 ± 0.2 | 0.3 ± 0.7 | 0.1 ± 0.2 | 0.26 |
| 13 | 1.9 | 2.7 | 4.5 | 3.9 ± 0.1 | 0.60 ± 0.02 | 0.16 | 2 | 2.8 | 4.6 | -1.2 ± 54.2 | 0.8 ± 7.6 | 5.0 ± 54.2 | 0.5 ± 1.1 | 0.16 |
| 14 | 1.9 | 2.6 | 4.3 | 3.9 ± 0.1 | 0.59 ± 0.02 | 0.14 | 1.9 | 2.7 | 4.4 | -0.1 ± 50.7 | 0.7 ± 79.7 | 3.9 ± 50.7 | 0.5 ± 1.2 | 0.15 |
| 15 | 1.6 | 2.3 | 3.9 | 3.5 ± 0.1 | 0.57 ± 0.02 | 0.10 | 1.6 | 2.3 | 4 | -0.5 ± 48.7 | 0.8 ± 15.5 | 4.0 ± 48.7 | 0.6 ± 1.1 | 0.11 |
| 16 | 1.8 | 2.5 | 4.2 | 3.8 ± 0.1 | 0.59 ± 0.02 | 0.12 | 1.7 | 2.5 | 4.5 | 1.9 ± 8.7 | 0.8 ± 1.6 | 2.0 ± 8.8 | 0.4 ± 0.6 | 0.18 |
| 17 | 2.1 | 2.9 | 4.8 | 4.1 ± 0.2 | 0.62 ± 0.02 | 0.17 | 2.2 | 3 | 5 | 0.5 ± 23.4 | 0.8 ± 11.1 | 3.6 ± 23.4 | 0.4 ± 0.7 | 0.21 |
| 18 | 2 | 2.8 | 4.6 | 4.1 ± 0.2 | 0.61 ± 0.02 | 0.17 | 2 | 2.8 | 4.7 | 1.4 ± 39.6 | 0.4 ± 2.6 | 2.7 ± 39.5 | 0.6 ± 2.5 | 0.25 |
| 19 | 1.5 | 2.2 | 3.8 | 3.6 ± 0.1 | 0.56 ± 0.02 | 0.10 | 1.5 | 2.2 | 4 | 0.9 ± 9.3 | 0.9 ± 4.0 | 2.7 ± 9.3 | 0.5 ± 0.5 | 0.14 |
| 20 | 2 | 2.8 | 4.6 | 4.1 ± 0.2 | 0.60 ± 0.02 | 0.17 | | | | | | | | |
| 21 | 1.6 | 2.3 | 3.9 | 3.7 ± 0.1 | 0.57 ± 0.01 | 0.09 | | | | | | | | |





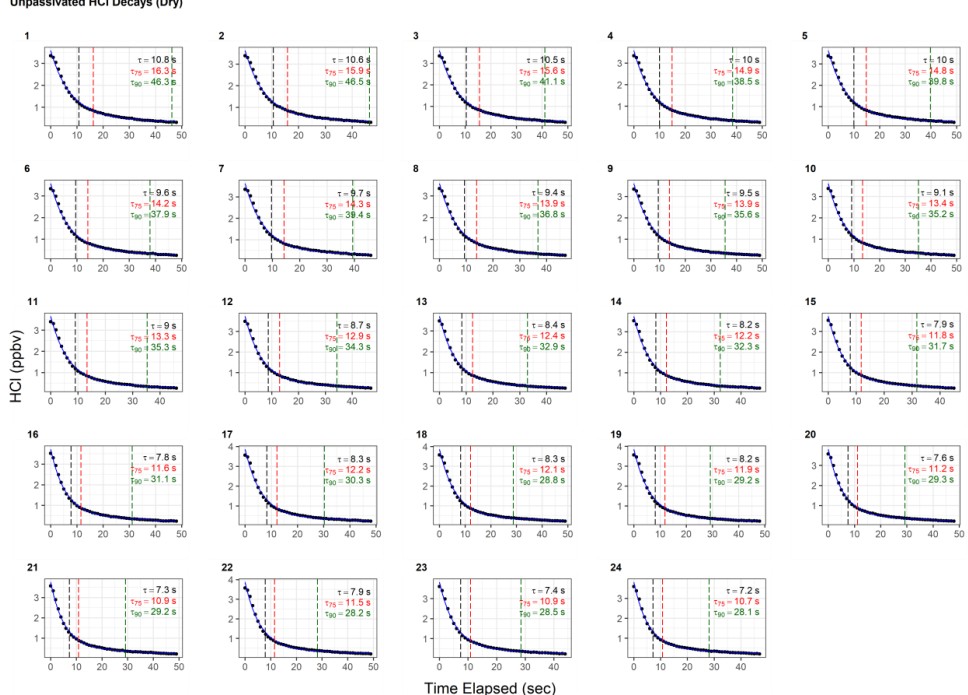


**Figure A2 – Instrument response times to changes in HCl mixing ratios without active chemical passivation. Black dots represent observed data and are overlayed by the calculated bi-exponential model (according to the terms listed in Table A2). Vertical hashed lines are placed on time elapsed corresponding to $\tau_e$ (black), $\tau_{75}$ (red), and $\tau_{90}$ (green).**




**Table A2 – Results for each model fit for determining the instrument response times without use of active chemical**
**passivation, using the 2.8 L min⁻¹ inertial inlet, and corresponds with Fig. A2.   Model parameters correspond to Eq. 1**
**in Sect. 2.6.2.**

| Trial | Single Exponential Fit | | | | | | Bi-Exponential Fit | | | | | | | |
| | $\tau_e$ (s) | $\tau_{75}$ (s) | $\tau_{90}$ (s) | $A_1$ | $k_1$ | Residuals | $\tau_e$ (s) | $\tau_{75}$ (s) | $\tau_{90}$ (s) | $A_1$ | $k_1$ | $A_2$ | $k_2$ | Residuals |
|---|---|---|---|---|---|---|---|---|---|---|---|---|---|---|---|
| 1 | 13.7 | 19.4 | 33 | $3.2 \pm 0.1$ | $0.935 \pm 0.003$ | 0.21 | 11.3 | 17.5 | | $2.9 \pm 0.1$ | $0.136 \pm 0.008$ | $0.7 \pm 0.1$ | $0.013 \pm 0.004$ | 0.06 |
| 2 | 13.5 | 19.1 | 32.4 | $3.2 \pm 0.1$ | $0.933 \pm 0.003$ | 0.21 | 11.1 | 17.1 | | $3.0 \pm 0.1$ | $0.139 \pm 0.009$ | $0.7 \pm 0.1$ | $0.012 \pm 0.004$ | 0.06 |
| 3 | 13 | 18.3 | 30.8 | $3.2 \pm 0.1$ | $0.930 \pm 0.003$ | 0.20 | 10.8 | 16.4 | | $3.0 \pm 0.1$ | $0.140 \pm 0.009$ | $0.7 \pm 0.1$ | $0.014 \pm 0.005$ | 0.06 |
| 4 | 12.7 | 17.9 | 30.3 | $3.2 \pm 0.1$ | $0.929 \pm 0.003$ | 0.20 | 10.5 | 16 | | $3.0 \pm 0.1$ | $0.142 \pm 0.008$ | $0.7 \pm 0.1$ | $0.014 \pm 0.004$ | 0.06 |
| 5 | 12.6 | 17.8 | 30.1 | $3.2 \pm 0.1$ | $0.928 \pm 0.003$ | 0.21 | 10.4 | 15.8 | | $3.0 \pm 0.1$ | $0.143 \pm 0.008$ | $0.7 \pm 0.1$ | $0.013 \pm 0.004$ | 0.06 |
| 6 | 12.4 | 17.6 | 29.8 | $3.2 \pm 0.1$ | $0.928 \pm 0.004$ | 0.22 | 10.1 | 15.5 | | $3.0 \pm 0.1$ | $0.148 \pm 0.008$ | $0.7 \pm 0.1$ | $0.013 \pm 0.004$ | 0.06 |
| 7 | 11.9 | 16.7 | 28.1 | $3.3 \pm 0.1$ | $0.923 \pm 0.004$ | 0.20 | 10 | 14.9 | | $3.1 \pm 0.1$ | $0.144 \pm 0.010$ | $0.6 \pm 0.1$ | $0.011 \pm 0.006$ | 0.07 |
| 8 | 11.4 | 16 | 27.1 | $3.2 \pm 0.1$ | $0.920 \pm 0.004$ | 0.19 | 9.6 | 14.3 | 42.2 | $3.0 \pm 0.1$ | $0.150 \pm 0.009$ | $0.6 \pm 0.1$ | $0.014 \pm 0.005$ | 0.06 |
| 9 | 11.6 | 16.2 | 27.3 | $3.2 \pm 0.1$ | $0.920 \pm 0.004$ | 0.20 | 9.7 | 14.5 | 43.1 | $3.1 \pm 0.1$ | $0.147 \pm 0.009$ | $0.6 \pm 0.1$ | $0.012 \pm 0.005$ | 0.07 |
| 10 | 11.3 | 16 | 27.1 | $3.2 \pm 0.1$ | $0.921 \pm 0.004$ | 0.21 | 9.4 | 14.1 | 43.1 | $3.0 \pm 0.1$ | $0.155 \pm 0.008$ | $0.6 \pm 0.1$ | $0.014 \pm 0.004$ | 0.06 |
| 11 | 11 | 15.5 | 26.1 | $3.3 \pm 0.1$ | $0.918 \pm 0.004$ | 0.21 | 9.2 | 13.8 | 42.1 | $3.1 \pm 0.1$ | $0.156 \pm 0.009$ | $0.6 \pm 0.1$ | $0.014 \pm 0.005$ | 0.06 |
| 12 | 10.7 | 15.1 | 25.5 | $3.3 \pm 0.1$ | $0.916 \pm 0.004$ | 0.21 | 8.9 | 13.4 | 39.4 | $3.1 \pm 0.1$ | $0.162 \pm 0.009$ | $0.6 \pm 0.1$ | $0.016 \pm 0.005$ | 0.06 |
| 13 | 10.5 | 14.9 | 25.3 | $3.3 \pm 0.1$ | $0.916 \pm 0.004$ | 0.22 | 8.6 | 13 | 39.7 | $3.1 \pm 0.1$ | $0.169 \pm 0.009$ | $0.7 \pm 0.1$ | $0.016 \pm 0.004$ | 0.06 |
| 14 | 10.3 | 14.6 | 24.9 | $3.3 \pm 0.1$ | $0.915 \pm 0.004$ | 0.22 | 8.4 | 12.8 | 38.9 | $3.1 \pm 0.1$ | $0.172 \pm 0.009$ | $0.7 \pm 0.1$ | $0.017 \pm 0.004$ | 0.06 |
| 15 | 9.9 | 14.1 | 24.1 | $3.3 \pm 0.1$ | $0.912 \pm 0.005$ | 0.22 | 8.1 | 12.3 | 37.8 | $3.1 \pm 0.1$ | $0.181 \pm 0.008$ | $0.7 \pm 0.1$ | $0.018 \pm 0.004$ | 0.05 |
| 16 | 9.9 | 14.2 | 24.3 | $3.3 \pm 0.1$ | $0.913 \pm 0.005$ | 0.22 | 8 | 12.2 | 37.6 | $3.1 \pm 0.1$ | $0.181 \pm 0.008$ | $0.7 \pm 0.1$ | $0.018 \pm 0.003$ | 0.05 |
| 17 | 10.2 | 14.3 | 24.1 | $3.5 \pm 0.1$ | $0.910 \pm 0.005$ | 0.22 | 8.6 | 12.7 | 36.9 | $3.3 \pm 0.1$ | $0.164 \pm 0.010$ | $0.6 \pm 0.1$ | $0.013 \pm 0.006$ | 0.08 |
| 18 | 10.1 | 14.2 | 23.9 | $3.5 \pm 0.1$ | $0.910 \pm 0.005$ | 0.22 | 8.5 | 12.6 | 35.4 | $3.3 \pm 0.1$ | $0.163 \pm 0.009$ | $0.6 \pm 0.1$ | $0.013 \pm 0.005$ | 0.07 |
| 19 | 10 | 14.1 | 23.7 | $3.5 \pm 0.1$ | $0.909 \pm 0.005$ | 0.22 | 8.4 | 12.4 | 35.8 | $3.3 \pm 0.1$ | $0.167 \pm 0.010$ | $0.6 \pm 0.1$ | $0.014 \pm 0.005$ | 0.07 |
| 20 | 9.6 | 13.6 | 23.1 | $3.4 \pm 0.1$ | $0.908 \pm 0.005$ | 0.23 | 7.8 | 11.7 | 36 | $3.2 \pm 0.1$ | $0.184 \pm 0.009$ | $0.7 \pm 0.1$ | $0.017 \pm 0.004$ | 0.06 |
| 21 | 9.3 | 13.4 | 23 | $3.3 \pm 0.1$ | $0.909 \pm 0.005$ | 0.23 | 7.5 | 11.4 | 35.4 | $3.1 \pm 0.1$ | $0.192 \pm 0.007$ | $0.7 \pm 0.1$ | $0.018 \pm 0.003$ | 0.05 |
| 22 | 9.7 | 13.7 | 23 | $3.5 \pm 0.1$ | $0.907 \pm 0.005$ | 0.23 | 8.1 | 12 | 34.4 | $3.3 \pm 0.1$ | $0.173 \pm 0.010$ | $0.6 \pm 0.1$ | $0.015 \pm 0.005$ | 0.08 |
| 23 | 9.2 | 13.1 | 22.2 | $3.4 \pm 0.1$ | $0.904 \pm 0.005$ | 0.22 | 7.6 | 11.4 | 34.2 | $3.2 \pm 0.1$ | $0.189 \pm 0.010$ | $0.7 \pm 0.1$ | $0.018 \pm 0.005$ | 0.06 |
| 24 | 9.1 | 13 | 22.1 | $3.4 \pm 0.1$ | $0.904 \pm 0.005$ | 0.23 | 7.4 | 11.2 | 34 | $3.2 \pm 0.1$ | $0.192 \pm 0.009$ | $0.7 \pm 0.1$ | $0.018 \pm 0.004$ | 0.06 |




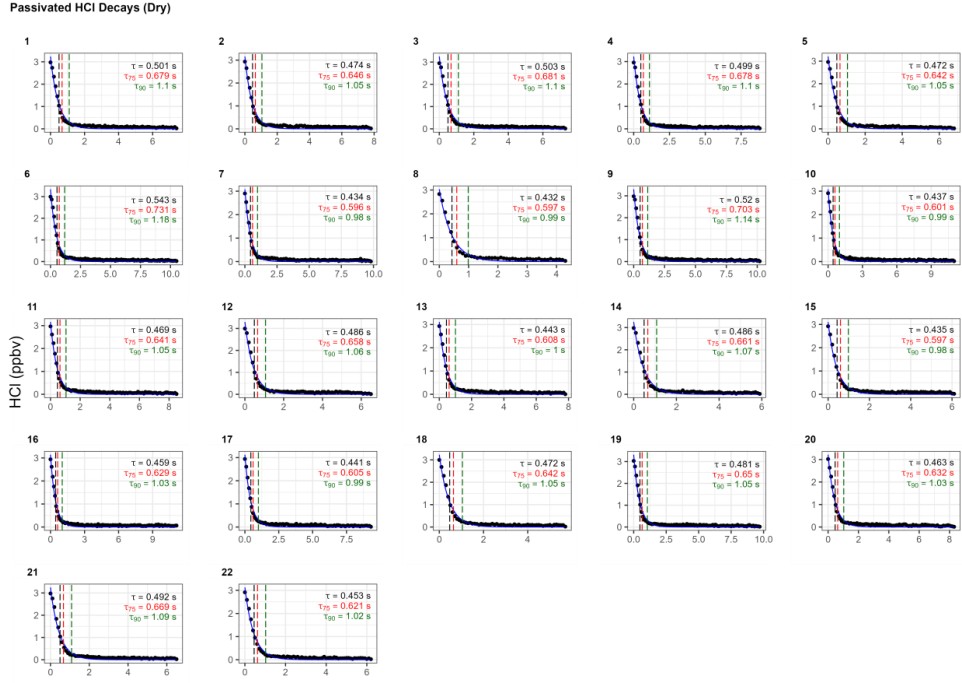

**Figure A3: Instrument response times to changes in HCl mixing ratios with active chemical passivation and a high flow inertial inlet (12.7 L min$^{-1}$). Black dots represent observed data and are overlayed by the calculated single exponential model (according to the terms listed in Table A3). Vertical hashed lines are placed on time elapsed corresponding to $\tau_e$ (black), $\tau_{75}$ (red), and $\tau_{90}$ (green).**


**Table A3: Results for each model fit for determining the instrument response times under actively passivated conditions with the 12.7 L min⁻¹ inertial inlet. Model parameters correspond to Eq. 1 in Sect. 2.6.2.**

| | | | | Single Exponential Fit | | |
|---|---|---|---|---|---|---|
| Trial | $\tau_e$ (s) | $\tau_{75}$ (s) | $\tau_{90}$ (s) | $A_1$ | $k_1$ | Residuals |
| 1 | 0.50 | 0.68 | 1.10 | $3.24 \pm 0.08$ | $0.113 \pm 0.009$ | 0.10 |
| 2 | 0.47 | 0.65 | 1.06 | $3.16 \pm 0.08$ | $0.106 \pm 0.009$ | 0.10 |
| 3 | 0.50 | 0.68 | 1.11 | $3.25 \pm 0.08$ | $0.12 \pm 0.01$ | 0.10 |
| 4 | 0.50 | 0.68 | 1.10 | $3.22 \pm 0.07$ | $0.115 \pm 0.009$ | 0.09 |
| 5 | 0.47 | 0.64 | 1.05 | $3.21 \pm 0.08$ | $0.103 \pm 0.009$ | 0.10 |
| 6 | 0.54 | 0.73 | 1.18 | $3.34 \pm 0.07$ | $0.129 \pm 0.009$ | 0.10 |
| 7 | 0.43 | 0.60 | 0.98 | $3.08 \pm 0.06$ | $0.093 \pm 0.007$ | 0.08 |
| 8 | 0.43 | 0.60 | 0.99 | $3.03 \pm 0.08$ | $0.10 \pm 0.01$ | 0.11 |
| 9 | 0.52 | 0.70 | 1.14 | $3.30 \pm 0.07$ | $0.121 \pm 0.009$ | 0.10 |
| 10 | 0.44 | 0.60 | 0.99 | $3.08 \pm 0.06$ | $0.095 \pm 0.007$ | 0.08 |
| 11 | 0.47 | 0.64 | 1.05 | $3.16 \pm 0.07$ | $0.106 \pm 0.008$ | 0.09 |
| 12 | 0.49 | 0.66 | 1.07 | $3.30 \pm 0.08$ | $0.105 \pm 0.009$ | 0.11 |
| 13 | 0.44 | 0.61 | 1.00 | $3.11 \pm 0.07$ | $0.097 \pm 0.008$ | 0.09 |
| 14 | 0.49 | 0.66 | 1.08 | $3.25 \pm 0.08$ | $0.109 \pm 0.009$ | 0.10 |
| 15 | 0.44 | 0.60 | 0.98 | $3.10 \pm 0.08$ | $0.092 \pm 0.008$ | 0.10 |
| 16 | 0.46 | 0.63 | 1.03 | $3.14 \pm 0.06$ | $0.103 \pm 0.007$ | 0.08 |
| 17 | 0.44 | 0.61 | 0.99 | $3.13 \pm 0.07$ | $0.095 \pm 0.008$ | 0.09 |
| 18 | 0.47 | 0.64 | 1.05 | $3.23 \pm 0.08$ | $0.104 \pm 0.009$ | 0.10 |
| 19 | 0.48 | 0.65 | 1.05 | $3.32 \pm 0.07$ | $0.102 \pm 0.008$ | 0.09 |
| 20 | 0.46 | 0.63 | 1.03 | $3.21 \pm 0.07$ | $0.101 \pm 0.008$ | 0.09 |
| 21 | 0.49 | 0.67 | 1.09 | $3.25 \pm 0.07$ | $0.112 \pm 0.009$ | 0.10 |
| 22 | 0.45 | 0.62 | 1.02 | $3.13 \pm 0.07$ | $0.101 \pm 0.008$ | 0.09 |

**Acknowledgements**

This program of work was primarily supported by the European Research Council (ERC-StG 802685). The Aerodyne Research Inc. HCl instrument development work was funded by the NOAA Small Business Innovation Research Program (WC-133R-17-CN-0092), and the Manchester field measurements were supported by the NERC SPF OSCA project (NE/T001917/1). The authors would also like to thank Abigail Mortimer, Stuart Murray, Chris Rhodes, and Mark Roper in the University of York Chemistry workshops, as well as Christopher Anthony, for technical support, and Stuart Lacy in the University of York Wolfson Atmospheric Chemistry Laboratory for data analysis software support. Further, the authors thank Michael Agnese and Michael Moore for TILDAS technical support. The authors would like to acknowledge the efforts Conner Daube made in testing configuration and sampling procedures on the companion HCl instrument. The author would also like to acknowledge the spectroscopic analysis performed by J. Barry McManus to diagnose non-ideal noise sources and design alignment optimizations. In addition, the authors thank James Lee, Will Drysdale, and Katie Read for their support in laboratory experiments involving $HNO_3$ quantification.





**Code availability**

The code used to perform the calculations used in this study will be made publicly available on completion of the review process. In the meantime, data can be obtained upon request from the corresponding author.

**Data availability**

The data used in this study will be made publicly available on completion of the review process. In the meantime, data can be obtained upon request from the corresponding author.

**Author contribution**

SCH, JRR, CD, and TIY designed, built, and tested the HCl TILDAS at Aerodyne Research, Inc. SSB and PRV were involved in the initial HCl detector testing and support of Aerodyne Research, Inc., instrument development. JWH and PME designed laboratory and field experiments, and JWH conducted laboratory and field experiments. SJA designed and constructed bespoke temperature controlling units for the inertial inlet, the field inlet box, and permeation source ovens. JS performed ISOROPPIA modelling experiments. MF provided $NO_x$ and $NO_y$ data, as well as provided critical field support during the OSCA campaign. JWH prepared the manuscript, and all authors reviewed the manuscript.

**Competing interests**

The authors declare that they have no conflict of interest.

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
