# Peer review of "Using Tunable Infrared Laser Direct Absorption Spectroscopy"

_Atmospheric Measurement Techniques, 2022_

## Author Response (AR1)

We thank all three Anonymous Referees for their careful reading of our manuscript and appreciate their thoughtful feedback. We have edited the manuscript to incorporate their critiques, which we feel has improved the overall quality of this work.

Below, individual Referee comments are presented in bold, and our responses are written in regular type. Line numbers in responses refer to the "track changes" version of the manuscript, which will be uploaded separately. For explicit clarity in text changes, added text will be in blue font, while deleted and/or moved text in  for clarity. Changes to figures will only be outlined in the responses to the Referee.

**Anonymous Referee 1**

**Comments to the Author:**

**The paper describes a commercial TILDAS instrument for measuring hydrogen chloride in ambient air and demonstrate the ability of sampling methodology to minimize inlet artefacts. Due to the "sticky" behavior of HCl gas, quantitative sampling remains a challenge for current approaches. To improve instrument response to changes in HCl gas concentration, a custom-fabricated quartz virtual impactor is used to replace particle filters to avoid excess surface-mediated interactions with filters, and the heating and PFBS coating methods are employed to improve transmission. Its performance validates that the sampling method is effective for reducing HCl "sticky" behavior. Overall, the paper is well written, with detailed characterization in the lab as well as reliable performance in the field sampling. I recommend this paper for publication in AMT after the following minor revisions.**

**General comments:**

**Section 2.2.1: The technique description of the TILDAS device is not clear, and more technical details need to be added, such as measurement principle, structural schematic diagram, etc.**

As pointed out by Anonymous Referees 2 and 3, the TILDAS technique is now many years old. Further detail is extensively given to these topics (including TILDAS measurement principle and structural schematic diagram) by McManus et al. (2011, 2015) for the TILDAS design used in this work:

McManus, J. B., Zahniser, M. S., and Nelson, D. D.: Dual quantum cascade laser trace gas instrument with astigmatic Herriott cell at high pass number, Appl. Opt., 50, A74, https://doi.org/10.1364/AO.50.000A74, 2011.

McManus, J. B., Zahniser, M. S., Nelson, D. D., Shorter, J. H., Herndon, S. C., Jervis, D., Agnese, M., McGovern, R., Yacovitch, T. I., and Roscioli, J. R.: Recent progress in laser-based trace gas instruments: performance and noise analysis, Appl. Phys. B, 119, 203–218, https://doi.org/10.1007/s00340-015-6033-0, 2015.

These publications are now properly cited, and the reader is now more clearly directed to these references for additional detail (lines 138-140 in below "track changes" manuscript). Additionally, we have included more details on the HCl specific attributes for the instrument used in this publication in Sect. 2.2.1 (lines 150-165).

**Section 2.2.2: The custom-fabricated quartz virtual impactor can effectively remove the large particles (> 300 nm diameter) in the sampling line, which was approximately 13% of the total volumetric flow. Only gas molecules and small particles (< 300 nm diameter) can flow into the TILDAS instrument. Please explain how does the impactor work and how is the ratio of flow rate obtained?**

The inertial inlet is interfaced with the instrument scroll pump, as seen in Fig. 1, which pulls air through the inertial inlet via two paths: 1) through a waste flowpath that does not pass through the TILDAS, and 2) through the TILDAS. Sample air that enters the inertial inlet is accelerated through a critical orifice into a low-pressure region (< 100 torr). Once in the low-pressure region, particulate separation occurs as follows: large particles (> 300 nm diameter) have large forward momentum and maintain their forward flow into the waste flow path (approximately 13% of the total volumetric flow, dictated by a separate critical orifice placed in the waste-flow path). Meanwhile, gas molecules and particles with an approximate diameter < 300 nm have less inertia and can make the 180° turn necessary to continue along the sample flow path through the TILDAS instrument (approximately 87% of the total volumetric flow). The resulting flow rate through the instrument was determined by the size of the critical orifice in the inertial inlet and cell pressure (set to approximately 40 torr).

We have revised this passage to clarify how particle separation occurs within the inertial inlet (lines 175 – 185). We have also referred the reader to Fig. 1 to visualize sample flow paths (line 187). Additionally, we have labeled the ambient pressure and low-pressure regions of the inertial inlet in Fig. 1.

**Section 3.1: The performance of HCl TILDAS is evaluated in the lab with dry zero air as well as in the field with HCl-scrubbed sample air, and its precision and LOD are superior to the previously reported methods. More technical details need to be added to explain how does the instrument achieve better performance? Did the authors perform long-term measurements of a fixed concentration of HCl gas? This approach can better represent its real performance.**

The better performance of the HCl TILDAS is achieved using a long pathlength (200 m), measuring absorptions in the mid-infrared by probing the fundamental ro-vibrational absorption band (which have a much larger cross-section than in the near-IR), and reducing light and dark noise levels to <5 x $10^{-6}$ equivalent absorbance in 1-second. We now provide these details in lines 310-313.

Regarding long-term measurements of a fixed concentration of a fixed gas, we performed a series of permeation source additions and removals across ~28 hours, resulting in 55, 10-min permeation source additions and subsequent 20-min background measurement periods. We note the permeation source concentration over this period was 4.1 ± 0.3 ppbv, and that permeation source concentration variability correlated closely with laboratory air-conditioning. Nevertheless, the average standard deviation calculated for the last five minutes of each permeation source additions was found to be 8 ± 2 pptv, while the average standard deviation of the last

five minutes of background periods was calculated as $7 \pm 1$ pptv, demonstrating nearly identical precisions while sampling blanks or fixed HCl concentrations.  These details have been added to Sect. 3.2.1, lines 356-360.

**Section 3.3.2: There is an obvious offset about 0.07 ppbv (shown in Figure 8) before addition of nitric acid to the passivated sample inlet flow. Please explain the reason for the offset signal.**

The data presented were not blank subtracted.  Figure 8 has been revised to use blank subtracted data.

**Section 3.4: The maximum concentration of HCl in field observation is about 0.1 ppbv shown in Figure 9(a). But the $HNO_3$ concentration of 4 ppbv may cause an increase of 0.08 ppbv of HCl. How to evaluate the error of atmospheric HCl concentration caused by $HNO_3$? And the influence of a potential leak on the measurement of HCl gas concentrations during observation needs to be clearly evaluated.**

The intention of the laboratory $HNO_3$ addition experiments (Sect. 3.3.2) was to demonstrate the potential of an acid-displacement-induced interference, as ambient sampling will be further complicated by additional strong acids, such as $H_2SO_4$, that may also cause rapid acid displacement reactions with HCl sorbed on inlet surfaces. As discussed in Sect. 3.3.2, the magnitude of the resulting HCl plume will be a function of how much HCl is taken up by instrument surfaces. This emphasizes the importance of reducing the amount of surface HCl available for off-gassing, though this effect will likely be dampened for in situ sampling, in which ambient strong acid concentrations change much more gradually.  We have added a comment to emphasize the importance of reducing HCl sorption (lines 491-492).

For the described field measurements (Sect. 3.4), the largest potential source of HCl for coating the inlet surfaces will be the regular permeation source or standardized HCl cylinder additions for 10 minutes every 3 hours.  Unfortunately, the absolute magnitude of sorbed HCl is difficult to quantify, as Fig. 7 demonstrates the complex relationship between stickiness and humidity, although regular measurement of inlet response times provides a metric by which it can be monitored, and the inlet subsequently cleaned if deemed an issue.  As discussed in Sect. 3.4, we do not see evidence of significant interference from $HNO_3$ (as estimated via $NO_z$) from our field results, given the differences in the diurnal profiles between HCl and $NO_z$.  Further, a preliminary comparison of data from the winter OSCA campaign shows virtually flat HCl signals while $NO_z$ maintains a diurnal profile, suggesting our results are likely not affected to a detectable degree by, at least, $HNO_3$ (please note that a 1 ppb offset has been added to the $NO_z$ data, and that the $NO_y$, $NO_2$, and NO data that were used to calculate $NO_z$ have not been QC-checked at the time of this response):

[Figure]

We additionally do not rule out future measurements from being affected by this potential interference. The dependence on inlet condition, however, means that we are able to estimate any additional uncertainty on the reported HCl using the regularly measured inlet response Tau values. Further, we are currently testing the implementation of temperature ramping the inlet to near 100 °C following additions of HCl standards to remove potential surface HCl from the inlet caused by the HCl standard addition, although a full field assessment of the impact of these temperature ramps has not yet been performed. We have added text to the Conclusions section about the potential improvements that this could have (lines 563-565).

Concerning the influence of a potential leak of $HNO_3$, we do not employ $HNO_3$ permeation sources during ambient HCl sampling, and therefore are not susceptible to such a leak.

**Specific comments:**

**Page 4, L137: The references could not be found in this manuscript.**

This has now been corrected, and these references have been properly added to the Reference section.

McManus, J. B., Zahniser, M. S., and Nelson, D. D.: Dual quantum cascade laser trace gas instrument with astigmatic Herriott cell at high pass number, Appl. Opt., 50, A74, https://doi.org/10.1364/AO.50.000A74, 2011.

McManus, J. B., Zahniser, M. S., Nelson, D. D., Shorter, J. H., Herndon, S. C., Jervis, D., Agnese, M., McGovern, R., Yacovitch, T. I., and Roscioli, J. R.: Recent progress in laser-based trace gas instruments: performance and noise analysis, Appl. Phys. B, 119, 203–218, https://doi.org/10.1007/s00340-015-6033-0, 2015.

**Page 14, L382-387: The influence of humidity on the measurement bias of HCl concentrations is only reported at 60% RH. In fact, the relative humidity of atmosphere is often much higher than this value. Therefore, the authors need to give the relationship between the measurement bias and the relative humidity, so that the reader can clearly grasp it.**

The pertinent passage in this comment refers to the effects of HCl standards being injected into the inlet under ambient relative humidities during the OSCA campaign. To that end, the mean and standard deviation presented were calculated for relative humidities between 60-93%, and compared with the mean and standard deviation of dry, compressed air at relative humidities below 20%. We have revised the sentence to clarify that the statistics and data presented in Fig 7b include these humidity ranges, and that the values are not solely obtained under an RH of 60% (lines 421-424).

**Page 18, L490: The data should be modified to 20 June 2021.**

The date has been modified (line 542).

**Anonymous Referee 2**

**Halfacre et al. report the construction and evaluation of a spectrometer for quantification of HCl in the atmosphere. The instrument is thoroughly described and was evaluated in the field as part of the Integrated Research Observation System for Clean Air" (OSCA) campaign in Manchester. The instrument's figures of merit are an improvement over existing technology (Table 1). The paper should be published once my comments below have been addressed.**

**Title: Please remove the term "Novelty" from the title. Novelty is implied when publishing. Further, TILDAS using astigmatic Herriott cells has been around for at least a quarter century. The main novelty of this work is the extension of known technology (QCL-TILDAS) to a new molecule (HCl).**

The title has been modified to "Using Tunable Infrared Laser Direct Absorption Spectroscopy for ambient hydrogen chloride detection: HCl-TILDAS".

**line 21/301 - "high accuracy". It would help to be more quantitative here and nuanced in the discussion of accuracy. The authors report that the instrument measured 3.6% lower than a commercial HCl cylinder, certified to contain a known concentration within ±5%. The certification applies to what the manufacturer added to the cylinder; what comes out can be an entirely different matter (subject to regulator passivation etc.). As such, a comparison to a single cylinder does not suffice to validate a new instrument's accuracy in my opinion.**

**In this context, are the absorption line strengths well known (and can be used to justify accuracy)?**

The line positions for HCl are extremely well known, ±0.0002 cm$^{-1}$ , with absorption cross-section line intensities uncertainties ranging between 1-2%:

Li, G., et al., Reference spectroscopic data for hydrogen halides, Part II: The line lists. Journal of Quantitative Spectroscopy and Radiative Transfer, 2013. 130: p. 284-295.

We have added this information and citation within Sect. 2.2.1 (lines 155-157). Further, the cylinder was not used as a calibration or correction tool, for the exact reasons the reviewer mentions, but instead as a validation of our method. As such, the slope / intercept reported via Fig 4 were not used to adjust any data (now clarified in Fig. 4 caption and lines 333-336). However, the combination of the well-known line-strengths and consistency of cylinder measurements are suitable for justifying the accuracy of this instrument.

**There is also a zero offset to be considered when discussing accuracy since the instrument reports negative mixing ratios (e.g., Figure 9) which are inaccurate by default. Consider stating a slope uncertainty and a zero offset uncertainty.**

The slope/uncertainty reported in Fig 4 were not used in correcting any data (now clarified in Fig. 4 caption, and lines 333-336). Concentrations were determined using the well-known absorption cross section line strengths, as above.

The negative mixing ratios shown in Fig. 9 are real measurements and would be expected for an instrument with a non-zero precision error measuring a concentration at or below its limit of detection. Higher time averaging of the data to reduce the random precision error results in HCl values within the stated uncertainty of zero for all time periods where negative mixing ratios are shown.

**Have the authors considered calibrating or comparing against a wet chemistry technique?**

We attempted to confirm our permeation device concentrations as measured on TILDAS via wet chemistry. Furlani et al (2021) describe a validation technique in which the permeation source is flowed into a basic solution over 24 hours, and then Cl$^-$ concentrations are confirmed via ion chromatography.

Furlani, T. C., Veres, P. R., Dawe, K. E. R., Neuman, J. A., Brown, S. S., VandenBoer, T. C., and Young, C. J.: Validation of a new cavity ring-down spectrometer for measuring tropospheric gaseous hydrogen chloride, Atmospheric Meas. Tech., 14, 5859–5871, https://doi.org/10.5194/amt-14-5859-2021, 2021.

However, our attempts to use this method with our standardized HCl cylinder produced inconsistent results, significantly underestimating the cylinder mixing ratio by at least a factor of 2. A variety of flow durations and conditions were attempted, but these experiments were ultimately abandoned since reproducibility could not be achieved.

**line 114 - replace detection with quantification**

The word "detection" has been replaced with "quantification".

**line 128 - I was wondering about the safety of perfluorobutanesulfonic acid, which is partially discussed on lines 186-188. Consider adding a comment regarding safe handling of this compound.**

We have now stated that PFBS is handled in a laboratory chemical fume hood, and that the bubbler that houses the chemical is installed within a sealed container that would contain any potential, albeit unlikely, spillage. This can now be found on lines 195-210.

**lines 147 - 154. Please state the line strengths (or cross-sections) used and how those were determined (Hitran?)**

Line strengths for all species are based upon the HITRAN 2016 database:

Gordon, I. E., et al.: The HITRAN2016 molecular spectroscopic database, J. Quant. Spectrosc. Radiat. Transf., 203, 3–69, https://doi.org/10.1016/j.jqsrt.2017.06.038, 2017.

 The HCl linestrength is 4.198 x $10^{-19}$ cm/molecule. This information has been included in Section 2.2.1 (lines 150-165.

**line 153 "well-resolved" - please state the FWHM of these lines and add a graph showing the spectrum you are discussing here (at high and low concentration), so the readers can see for themselves.**

The FWHM is 0.010 cm-1, which is primarily pressure- and doppler-broadened. The inherent laser linewidth is <0.001 cm-1.  This information has been added lines 150-165.  Additionally, the requested HITRAN spectrum has been added as Fig. A1.

**line 501-514. Please try to be more quantitative in this paragraph - for example, rather than saying 'greatly improve' or 'higher flow inlets' or 'reduce sample air residence time', state by how much or the actual value.**

This section has been amended to include pertinent quantitative details (lines 555-570).

**Figure 6 - replace sec with s**

This has been corrected.

**Figure 8 - what caused the second hump at 14:10? Please add an explanation to the caption.**

It was not determined what caused the $2^{nd}$ hump.  These experiments occurred in an air-conditioned laboratory, so it is possible it is related to an ambient temperature effect on the instrument inlet or $HNO_3$ permeation device. However, this experiment was repeated several times, and we have replaced this figure with an alternate experiment where a $2^{nd}$ hump is not present.

**Figure 9 panels (a) and (c) - both the HCl and NO2 data exhibit spikes and negative concentrations, even when averaged. Please add some discussion to the text as to the meaning of this, potential causes, and remedies. Consider adding a horizontal line to show limits of detection or quantification.**

Negative concentrations for the reported HCl data may be caused by blank measurements being recorded as larger than ambient measurements. Cylinder HCl addition experiments during the day between 19-21 July may have influenced the blanks if not enough time for signal recovery was allowed. This has been clarified in the caption of Fig. 9. We have additionally added a 1Hz, 3 sigma LOD line to Fig. 9.

Despite calibrations and nearly complete convertor efficiency testing, the $NO_z$ measurement is highly uncertain, as its calculation relies on individual measurements of $NO_y$, NO, and $NO_2$; because it is an additive measurement, the uncertainties are also additive. This has been explored in more detail by San Martini et al. (2006):

San Martini, F. M., Dunlea, E. J., Grutter, M., Onasch, T. B., Jayne, J. T., Canagaratna, M. R., Worsnop, D. R., Kolb, C. E., Shorter, J. H., Herndon, S. C., Zahniser, M. S., Ortega, J. M., McRae, G. J., Molina, L. T., and Molina, M. J.: Implementation of a Markov Chain Monte Carlo method to inorganic aerosol modeling of observations from the MCMA-2003 campaign – Part I: Model description and application to the La Merced site, Atmospheric Chem. Phys., 6, 4867–4888, https://doi.org/10.5194/acp-6-4867-2006, 2006.

We now emphasize the high uncertainty in this $NO_z$ measurement and that it is used for comparative, not quantitative, purposes (lines 517-519).

**Figure 10 panel (a) - change units to pptv to avoid the x10³**

This figure has been revised to avoid the $x10^3$.

**Anonymous Referee 3**

**The manuscript describes a gas phase HCL sensor, based on tunable infrared laser direct absorption spectroscopy (TILDAS). The authors highlight the importance of HCL in the atmosphere, describing its influence. They describe the difficulties of monitoring HCL, it is "sticky" and results in such effects as long instrument response times. They describe current monitoring techniques, their strengths and weaknesses, and why their approach will be of benefit.**

**The authors first describe the TILDAS sensor design, followed by techniques to minimize sticky behavior of HCL on extraction, an "inertial inlet" and active passivation. They also describe procedures for validation, field testing and data analysis.**

**The authors present results for different configurations, such as with and without passivation and humidity effects, They present field data and compare sensitivity to other published set-ups. They discuss problems with HCL particulate and nitric acid.**

**They present 7-8 pptv at 1 Hz and 3ð• œŽ limit of detection ranging from 21-24 pptv. For longer averaging times, the highest precision obtained was 0.5 pptv and 3ð• œŽ limit of detection of 1.6 pptv at 2.4 minutes. These values are competitive compared to other optical techniques, which are considered more complicated to set-up. I think the manuscript should be published.**

**I believe the manuscript requires minor revisions and clarifications.**

- **The title needs to be considered. I find it misleading. TILDAS is not a novel spectroscopic approach. It is the first application of TILDAS to HCL. Should be clarified to reader or manuscript changed.**

  The title has been modified to "Using Tunable Infrared Laser Direct Absorption Spectroscopy for ambient hydrogen chloride detection: HCl-TILDAS".

- **There is little detail on optical configuration of set-up. If it is new and custom made, more information can be given here.**

  As pointed out by the Referee, the TILDAS technique is now many years old. Further detail is extensively given to these topics (including TILDAS measurement principle and structural schematic diagram) by McManus et al. (2011, 2015) for the TILDAS design used in this work:

  McManus, J. B., Zahniser, M. S., and Nelson, D. D.: Dual quantum cascade laser trace gas instrument with astigmatic Herriott cell at high pass number, Appl. Opt., 50, A74, https://doi.org/10.1364/AO.50.000A74, 2011.

  McManus, J. B., Zahniser, M. S., Nelson, D. D., Shorter, J. H., Herndon, S. C., Jervis, D., Agnese, M., McGovern, R., Yacovitch, T. I., and Roscioli, J. R.: Recent progress in laser-based trace gas instruments: performance and noise analysis, Appl. Phys. B, 119, 203–218, https://doi.org/10.1007/s00340-015-6033-0, 2015.

  These publications are now properly cited, and the reader is now more clearly directed these references for additional detail (lines 137-140). Additionally, we have included more details on the HCl specific attributes for the instrument used in this publication in Sect. 2.2.1 (lines 150-165).

- **There is little detail to spectral fitting. They do talk about background subtration. But, error can also come from the spectra fit. I think more detail should be given here.**

Spectral fits are non-linear least squares fits of a ~1 cm$^{-1}$ spectral window, using a nonlinear least-squares fit that includes a polynomial baseline. Peak location is fixed using a frequency-locking algorithm based upon the deep methane lines. Pressure and temperature are included in the fit to account for pressure broadening and rovibrational state populations, respectively. These details have been added to Sect. 2.2.1 (lines 150-165).

- **Re Methane measurement. Do you have a LOD or senitivity for this measurement? it appears in plot, methane can fluctuate by approx 2-3 ppb in a few seconds. is this real? If yes, why not see these typeos of fluctuations with HCL**

The line strengths used for the methane measurement are approximately 10x less than for HCl, and is therefore expected to be a less precise measurement. For the data presented in Fig. 6, standard deviations for the high concentration average 3 ppbv, and 2 ppbv while sampling the lower concentrations. This would result in an approximate LOD of 4-9 ppbv. However, it should also be clarified that the methane data presented in Fig. 6 for the submitted manuscript used normalized concentrations that represent a change in methane from a high concentration from a zero-air cylinder (~2250 ppbv) to a lower concentration as measured by ambient air (~2220 ppbv); the normalization was done for timescale of signal decay calculation comparisons. However, we have modified Fig. 6 to restore the methane mixing ratios as originally observed, and to make it clear to the reader we are not operating near these estimated limits of detection. Further characterization of this methane signal is outside the scope of this paper.

- **ISORROPIA II.. a few sentences on theory are not in referred section**

This has been corrected; the appropriate references now appear in the References section.

- **The conclusion seems more like an outlook, except for first sentence.**

More summary detail has been added to the Conclusion section (lines 555-570).

**Individual line comments/typos;**

**line 107... typo "it is has"**

This has been corrected (line 108).

**line 138... What is exact wavelength of laser and tuning range?**

Laser radiation probes the strong R(1) H$^{35}$Cl line (2925.89645 cm$^{-1}$) of the (1-0) rovibrational absorption band near 3.4 $\mu$m (lines 154-155). The spectral window (2925.80 to 2926.75 cm$^{-1}$) has been added to Sect. 2.2.1 (lines 150-153).

**line 160... "gas phase via acid displacement" can you add a reference for this. There are references earlier, but they don;t seem to fit this.**

We have added a reference to Roscioli et al., 2016 which demonstrates this effect for nitric acid. We have also clarified that we are referring to a mechanism analogous to that which occurs with aerosol (e.g, Beichert and Finlayson-Pitts, 1997), line 172.

Roscioli, J. R., Zahniser, M. S., Nelson, D. D., Herndon, S. C., and Kolb, C. E.: New Approaches to Measuring Sticky Molecules: Improvement of Instrumental Response Times Using Active Passivation, J. Phys. Chem. A, 120, 1347–1357, https://doi.org/10.1021/acs.jpca.5b04395, 2016.
Beichert, P. and Finlayson-Pitts, B. J.: Knudsen Cell Studies of the Uptake of Gaseous HNO3 and Other Oxides of Nitrogen on Solid NaCl: The Role of Surface-Adsorbed Water, J. Phys. Chem., 100, 15218–15228, https://doi.org/10.1021/jp960925u, 1996.

**line 401 "It is well established that HCl and particulate chloride (pCl- 401 ) exist together in dynamic equilibrium" can you add refernce here.**

We have inserted citations for the following references (lines 440-442):

Beichert, P. and Finlayson-Pitts, B. J.: Knudsen Cell Studies of the Uptake of Gaseous HNO3 and Other Oxides of Nitrogen on Solid NaCl: The Role of Surface-Adsorbed Water, J. Phys. Chem., 100, 15218–15228, https://doi.org/10.1021/jp960925u, 1996.
Brimblecombe, P. and Clegg, S. L.: The solubility and behaviour of acid gases in the marine aerosol, J. Atmospheric Chem., 7, 1–18, https://doi.org/10.1007/BF00048251, 1988.
Clegg, S. L. and Brimblecombe, P.: The dissociation constant and henry's law constant of HCl in aqueous solution, Atmospheric Environ. 1967, 20, 2483–2485, https://doi.org/10.1016/0004-6981(86)90079-X, 1986.
Fountoukis, C. and Nenes, A.: ISORROPIA II: a computationally efficient thermodynamic equilibrium model for $K^+$-$Ca^{2+}$-$Mg^{2+}$-$NH_4^+$-$Na^+$-$SO_4^{2-}$-$NO_3^-$-$Cl^-$$H_2O$ aerosols, Atmospheric Chem. Phys., 7, 4639–4659, https://doi.org/10.5194/acp-7-4639-2007, 2007.

**line 484 How is nh3 measuremed here?**

A Los Gatos Research ammonia analyzer was used. This is now explicitly stated on line 536.

**Figure 10. Is the time on a) and b) the same. Not clear. y scale strange**

The timescale on a) and b) are the same. Vertical guidelines have been added to make this clear. The y-axis labels have been modified.